# LIN37-DREAM prevents DNA end resection and homologous recombination at DNA double-strand breaks in quiescent cells

Bo-Ruei Chen[1,2], Yinan Wang[3], Anthony Tubbs[4], Dali Zong[4], Faith C Fowler[3], Nicholas Zolnerowich[4], Wei Wu[4], Amelia Bennett[3], Chun-Chin Chen[3], Wendy Feng[1], Andre Nussenzweig[4], Jessica K Tyler[3]*, Barry P Sleckman[1,2]*

[1]Division of Hematology and Oncology, Department of Medicine, University of Alabama at Birmingham, Birmingham, United States; [2]O'Neal Comprehensive Cancer Center, University of Alabama at Birmingham, Birmingham, United States; [3]Department of Pathology and Laboratory Medicine, Weill Cornell Medicine, New York, United States; [4]Laboratory of Genome Integrity, National Cancer Institute, Bethesda, United States

**Abstract** DNA double-strand break (DSB) repair by homologous recombination (HR) is thought to be restricted to the S- and $G_2$- phases of the cell cycle in part due to 53BP1 antagonizing DNA end resection in $G_1$-phase and non-cycling quiescent ($G_0$) cells. Here, we show that LIN37, a component of the DREAM transcriptional repressor, functions in a 53BP1-independent manner to prevent DNA end resection and HR in $G_0$ cells. Loss of LIN37 leads to the expression of HR proteins, including BRCA1, BRCA2, PALB2, and RAD51, and promotes DNA end resection in $G_0$ cells even in the presence of 53BP1. In contrast to 53BP1-deficiency, DNA end resection in LIN37-deficient $G_0$ cells depends on BRCA1 and leads to RAD51 filament formation and HR. LIN37 is not required to protect DNA ends in cycling cells at $G_1$-phase. Thus, LIN37 regulates a novel 53BP1-independent cell phase-specific DNA end protection pathway that functions uniquely in quiescent cells.

*For correspondence:
jet2021@med.cornell.edu (JKT);
bps@uab.edu (BPS)

## Introduction

DNA double-strand breaks (DSBs) are repaired by two main pathways, non-homologous end joining (NHEJ) and homologous recombination (HR) (*Prakash et al., 2015*; *Chang et al., 2017*). HR functions in the S- and $G_2$-phases of the cell cycle using the sister chromatid as a template for precise homology-directed repair of DSBs. In contrast, NHEJ functions in all phases of the cell cycle to rejoin broken DNA ends and is the only pathway of DSB repair in $G_1$-phase cells and in non-cycling cells that have exited the cell cycle and are quiescent in $G_0$, which comprise the majority of cells in the human body (*Chang et al., 2017*). The initiation of HR requires DNA end resection to generate extended singlestranded DNA (ssDNA) overhangs that are coated by the trimeric ssDNA binding protein complex RPA1/2/3 (hereafter referred to as RPA) (*Ciccia and Elledge, 2010*). RPA is subsequently replaced by RAD51, and the RAD51 nucleofilament mediates a search for a homologous template usually within the sister chromatid to enable the completion of HR-mediated DNA DSB repair (*Mimitou and Symington, 2009*; *Wyman et al., 2004*). In contrast, NHEJ works best on DNA ends with minimal ssDNA overhangs, necessitating that DNA end resection be limited in $G_1$-phase and $G_0$ cells (*Chang et al., 2017*; *Symington and Gautier, 2011*). Extensive resection and ssDNA generation at broken DNA ends in cells at $G_0/G_1$ would antagonize NHEJ-mediated DSB repair and

promote aberrant homology-mediated repair, due to the absence of sister chromatids, forming chromosomal translocations and deletions leading to genome instability (*Ciccia and Elledge, 2010*). Therefore, the generation of ssDNA at broken DNA ends is the critical decision point for whether a DSB will be repaired by NHEJ and HR. As such, highly regulated processes that control DNA end resection are critical for ensuring the appropriate choice of DSB repair pathways in all cells.

During HR, BRCA1 initiates the resection of broken DNA ends with the CtIP and MRE11 nucleases generating short ssDNA tracts that are extended by other nucleases such as EXO1 and DNA2-BLM (*Symington and Gautier, 2011*). Other proteins, including Fanconia anemia (FA) proteins such as FANCD2, are also involved in regulating DNA end resection (*Unno et al., 2014*; *Murina et al., 2014*; *Cai et al., 2020*). In cells where NHEJ must repair DSBs, the extensive resection of DNA ends must be prevented and 53BP1 and its downstream effectors RIF1 and the shieldin complex antagonize DNA end resection in these cells (*Setiaputra and Durocher, 2019*; *Mirman and de Lange, 2020*; *Bunting et al., 2010*). 53BP1 and shieldin may protect DNA ends by inhibiting the recruitment or activity of pro-resection proteins and also by promoting DNA synthesis at resected DNA ends to 'fill in' the ssDNA gap generated by resection (*Mirman and de Lange, 2020*; *Setiaputra and Durocher, 2019*).

Whether pathways exist in addition to 53BP1 and RIF1/shieldin that protect DNA ends and promote genome integrity by preventing aberrant homology-mediated DSB repair is unclear. Here, we develop an unbiased whole-genome CRISPR/Cas9 screening approach based on assaying RPA loading at DSBs to identify genes encoding proteins that prevent DNA end resection and ssDNA generation in cells with 2N DNA content, which will be in either $G_0$ or $G_1$. This approach identified guide RNAs (gRNAs) targeting genes encoding proteins known to protect DNA ends such as 53BP1 and RIF1. However, this screen also identified LIN37, a protein not known to function in DNA end processing and DSB repair. LIN37 is a component of the DREAM (dimerization partner, RB-like, E2F, and multi-vulval) transcriptional repressor complex (*Litovchick et al., 2007*; *Sadasivam and DeCaprio, 2013*; *Mages et al., 2017*). Here, we show that LIN37 functions in a 53BP1-independent manner to protect DNA ends from resection exclusively in quiescent $G_0$ cells. Moreover, while loss of either 53BP1 or LIN37 leads to loading of RPA at DSBs in $G_0$ cells, loss of LIN37 further leads to subsequent steps of HR including RAD51 loading and homology-mediated DSB repair.

## Results

### Identification of novel factors that regulate DNA end resection

Murine pre-B cells transformed with the Abelson murine leukemia virus kinase, hereafter referred to as abl pre-B cells, are rapidly cycling cells (*Rosenberg et al., 1975*). When treated with the abl kinase inhibitor, imatinib, abl pre-B cells stop cycling with a predominantly 2N DNA content indicative of them being in $G_0/G_1$ (*Bredemeyer et al., 2006*; *Muljo and Schlissel, 2003*). Imatinib treated abl pre-B cells that ectopically express Bcl2 survive in culture for several days and we refer to these cells as non-cycling abl pre-B cells (*Bredemeyer et al., 2006*).

DNA DSBs in $G_0/G_1$ cells are repaired by NHEJ, which requires DNA Ligase 4 to join the broken DNA ends (*Chang et al., 2017*). Consequently, DNA DSBs generated in non-cycling DNA Ligase 4-deficient ($Lig4^{-/-}$) abl pre-B cells persist un-repaired (*Helmink et al., 2011*). These persistent DSBs are protected from nucleolytic activity and exhibit minimal DNA end resection (*Tubbs et al., 2014*; *Dorsett et al., 2014*; *Helmink et al., 2011*). However, loss of H2AX or 53BP1 in abl pre-B cells leads to extensive DNA end resection and the generation of ssDNA overhangs (*Tubbs et al., 2014*; *Dorsett et al., 2014*; *Helmink et al., 2011*). ssDNA at resected DNA ends binds to RPA, which can be assayed by flow cytometry after detergent extraction of soluble RPA (*Forment et al., 2012*). Indeed, as compared to non-cycling $Lig4^{-/-}$ abl pre-B cells, the generation of DSBs by ionizing radiation (IR) in non-cycling $Lig4^{-/-}:Trp53bp1^{-/-}$ abl pre-B cells led to robust chromatin-associated RPA as assayed by flow cytometry (*Figure 1A*). This assay detects more extensive DNA end resection in the absence of 53BP1 and thus serves as the basis for a screen to identify additional proteins that normally function to prevent extensive DNA end resection in $G_0/G_1$ cells.

To facilitate our screen for gRNAs that increase the association of RPA with DSBs, we introduced a lentivirus containing a FLAG-Cas9 cDNA under the control of a doxycycline-inducible promoter into $Lig4^{-/-}$ abl pre-B cells. Immunodetection of the FLAG epitope allowed flow cytometric

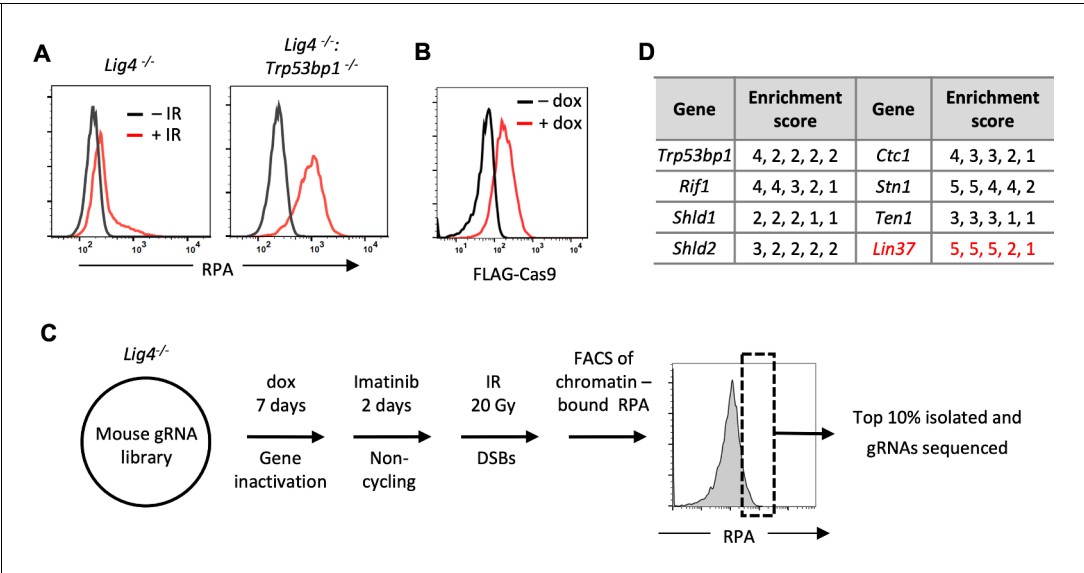

**Figure 1.** An unbiased genome-scale gRNA screen for novel DNA end protection factors. (**A**) Flow cytometric analysis of chromatin-bound RPA before and after IR of non-cycling *Lig4*$^{-/-}$ and *Lig4*$^{-/-}$:*Trp53bp1*$^{-/-}$ abl pre-B cells. (**B**) Flow cytometric analysis of FLAG-Cas9 in *Lig4*$^{-/-}$ cells with (+dox) and without (−dox) doxycycline to induce expression of FLAG-Cas9. (**C**) Schematic diagram of the genome-scale guide RNA screen for genes preventing DNA end resection in non-cycling *Lig4*$^{-/-}$ abl pre-B cells. (**D**) Enrichment score (fold enrichment) of individual gRNAs to a subset of genes identified in the RPA high population. gRNA, guide RNA; IR, ionizing radiation.

The online version of this article includes the following source data for figure 1:

**Source data 1.** RPA screen result in non-cycling *Lig4*$^{-/-}$ abl pre-B cells.

assessment of FLAG-Cas9 protein expression upon doxycycline treatment, showing that FLAG-Cas9 can be induced in all the cells in the population (**Figure 1B**). A lentiviral mouse gRNA library containing approximately 90,000 gRNAs to 18,000 mouse genes was introduced into these cells followed by doxycycline treatment for 7 days to promote Cas9-gRNA-mediated gene inactivation (**Figure 1C**; *Tzelepis et al., 2016*). After 2 days of imatinib treatment to render these cells non-cycling, the cells were subjected to irradiation (IR) and those with the highest level (top 10%) of chromatin-bound RPA were isolated by flow cytometric cell sorting (**Figure 1C**). The gRNAs from this 'RPA-high' cell population were sequenced, and the frequency of individual gRNAs in the RPA-high population relative to those in the 'RPA-low' population was determined.

Multiple gRNAs to genes encoding proteins known to prevent DNA end resection such as 53BP1, RIF1, and shieldin subunits were enriched in the RPA-high cell population, demonstrating the veracity of this screening approach (**Figure 1D** and **Figure 1—source data 1**). In addition, several gRNAs to *Lin37*, a gene encoding the LIN37 subunit of the MuvB complex, were also highly enriched in RPA-high cells (**Figure 1D** and **Figure 1—source data 1**; *Litovchick et al., 2007*). When bound to the pocket proteins p130/p107, and E2F proteins E2F4/5 and DP, the MuvB complex forms the DREAM transcription repressor complex, which functions to repress genes that promote entry into the cell cycle (*Litovchick et al., 2007*; *Sadasivam and DeCaprio, 2013*; *Mages et al., 2017*).

## RPA accumulation at DNA DSBs in non-cycling LIN37-deficient cells

We generated DNA Ligase 4-deficient abl pre-B cells that were deficient in LIN37 (*Lig4*-$^{-/-}$:*Lin37*$^{-/-}$) (**Figure 2A**). Similar to 53BP1-deficient abl pre-B cells, LIN37-deficient cells also exhibited increased RPA association with DSBs in chromatin after IR, with most of the cells lacking LIN37 having even more chromatin-bound RPA after IR than cells lacking 53BP1 (**Figure 2B**). This is not a consequence of increased IR-induced DNA damage in the absence of LIN37, because all the cells had similar levels of γH2AX indicative of similar DNA DSB generation in response to IR (**Figure 2B**). Analyses of non-cycling wild type (WT), *Trp53bp1*$^{-/-}$, and *Lin37*$^{-/-}$ abl pre-B cells that expressed Ligase 4 yielded similar results, indicating that RPA binding at DSBs in 53BP1 and LIN37 deficient abl pre-B cells does not depend on DNA Ligase 4 deficiency (**Figure 2—figure supplement 1**). After irradiation, RPA

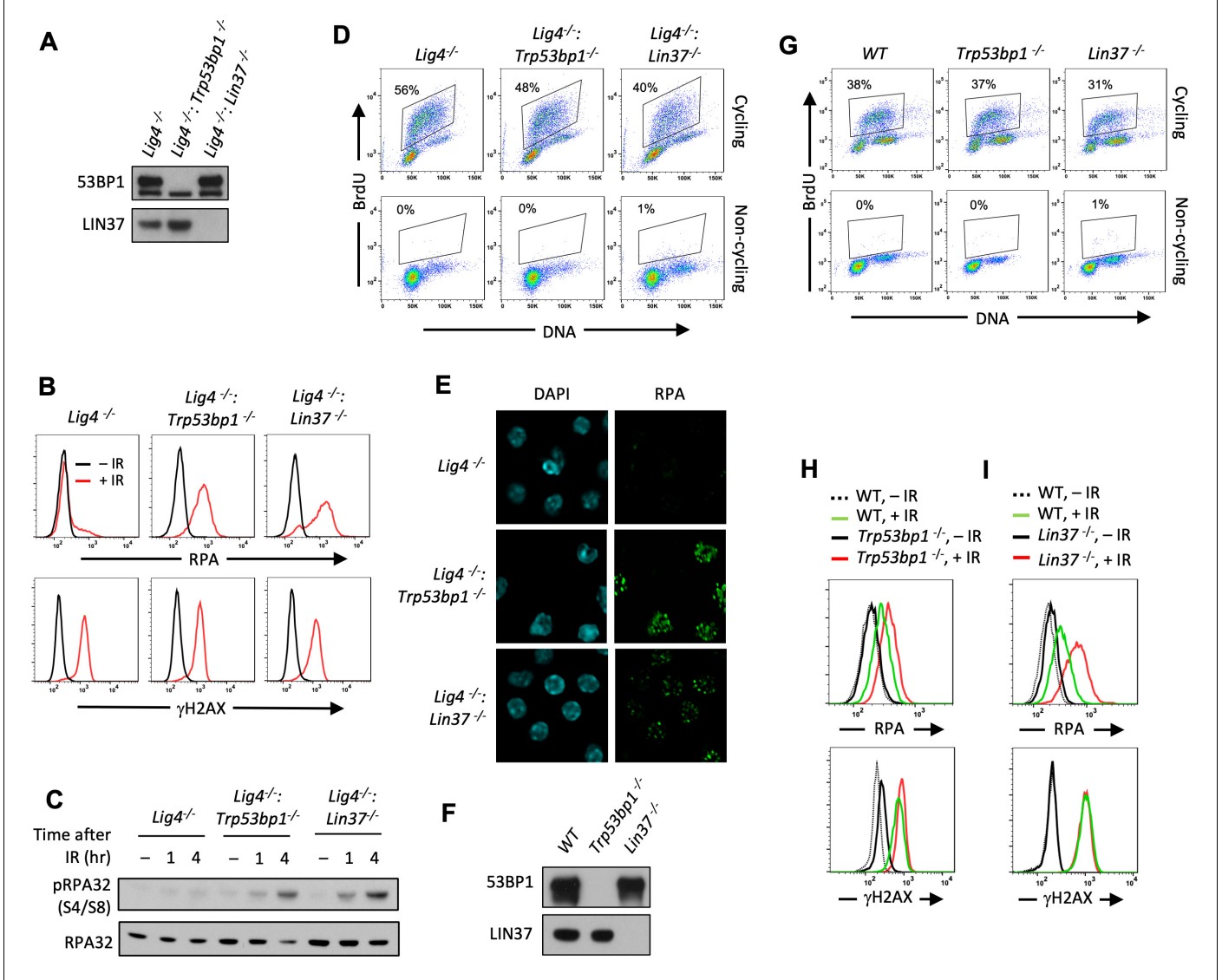

**Figure 2.** Non-cycling LIN37-deficient cells accumulate chromatin-bound RPA after IR-induced damage. (A) Western blot analysis of indicated proteins in $Lig4^{-/-}$, $Lig4^{-/-}:Trp53bp1^{-/-}$, and $Lig4^{-/-}:Lin37^{-/-}$ abl pre-B cells. (B) Flow cytometric analysis of chromatin-bound RPA (top) and γH2AX (bottom) before and after IR of non-cycling $Lig4^{-/-}$, $Lig4^{-/-}:Trp53bp1^{-/-}$, and $Lig4^{-/-}:Lin37^{-/-}$ abl pre-B cells. The experiments were repeated in two independently generated cell lines at least twice. (C) Western blot analysis of RPA32 and phosphorylated RPA 32 (pRPA32(S4/S8)) of non-cycling $Lig4^{-/-}$, $Lig4^{-/-}:Trp53bp1^{-/-}$, and $Lig4^{-/-}:Lin37^{-/-}$ abl pre-B cells treated without or with IR after indicated times. (D) Flow cytometric analysis of cycling and non-cycling $Lig4^{-/-}$, $Lig4^{-/-}:Trp53bp1^{-/-}$, and $Lig4^{-/-}:Lin37^{-/-}$ abl pre-B cells for BrdU incorporation and DNA content (7-AAD). Percentage of cells in S-phase is indicated. (E) Representative images of IR-induced RPA foci in non-cycling $Lig4^{-/-}$, $Lig4^{-/-}:Trp53bp1^{-/-}$, and $Lig4^{-/-}:Lin37^{-/-}$ abl pre-B cells from two independent experiments. (F) Western blot analysis of indicated proteins in WT, $Trp53bp1^{-/-}$, and $Lin37^{-/-}$ MCF10A cells. (G) Flow cytometric analysis of BrdU pulsed cycling (top) or non-cycling (bottom) WT, $Trp53bp1^{-/-}$, and $Lin37^{-/-}$ MCF10A cells as in (D). (H, I) Flow cytometric analysis of chromatin-bound RPA (top) and γH2AX (bottom) before or after IR of non-cycling WT and (H) $Trp53bp1^{-/-}$ or (I) $Lin37^{-/-}$ MCF10A cells. IR, ionizing radiation; WT, wild type.

The online version of this article includes the following figure supplement(s) for figure 2:

**Figure supplement 1.** Non-cycling LIN37-deficient cells accumulate chromatin-bound RPA after IR-induced damage.

phosphorylation at serine 4 and serine 8 of the RPA32 subunit was also detected at higher levels in non-dividing $Lig4^{-/-}:Trp53bp1^{-/-}$ and $Lig4^{-/-}:Lin37^{-/-}$ as compared to $Lig4^{-/-}$ abl pre-B cells, further suggesting that this RPA binding is at resected DNA ends (*Figure 2C*; *Maréchal and Zou, 2015*).

LIN37 is a component of the DREAM complex, which along with RB prevents cells from entering S-phase where resection of DNA ends and RPA loading normally occur at DSBs as part of HR (*Weinberg, 1995*; *Sadasivam and DeCaprio, 2013*; *Ciccia and Elledge, 2010*). However, loss of LIN37 does not cause imatinib-treated abl pre-B cells to enter S-phase as evidenced by imatinib-treated *Lig4$^{-/-}$:Lin37$^{-/-}$* abl pre-B cells having primarily 2N DNA content and not incorporating BrdU, which is indicative of DNA synthesis in S-phase cells (*Figure 2D*). If RPA is binding to chromatin at DSBs this binding should form discrete RPA IR-induced foci (IRIF) that can be visualized by immunofluorescence staining. Indeed, chromatin-bound RPA in IR treated non-cycling *Lig4$^{-/-}$:Trp53bp1$^{-/-}$* and *Lig4$^{-/-}$:Lin37$^{-/-}$* abl pre-B cells formed discrete nuclear foci indicative of localization at DSBs (*Figure 2E*).

To evaluate LIN37 function in a different cell type using a distinct approach to render them non-cycling, we generated LIN37- and 53BP1-deficient MCF10A human mammary epithelial cells. Upon withdrawal of epidermal growth factor (EGF) these cells stop cycling, have 2N DNA content, and lack of BrdU incorporation consistent with the notion that they are in G$_0$/G$_1$ (*Figure 2F and G*). Non-cycling *Lin37$^{-/-}$* and *Trp53bp1$^{-/-}$* MCF10A cells also exhibited increased chromatin-bound RPA after IR as compared to WT MCF10A cells (*Figure 2H and I*), indicating that the function of LIN37 in suppressing RPA accumulation at DNA DSBs in non-cycling cells is conserved in human and mouse. As was seen above in murine cells, the extent of RPA accumulation in human cells lacking LIN37 after IR was greater than seen in cells lacking 53BP1, suggesting that LIN37 plays an important role in preventing DNA end resection and extensive generation of ssDNA at DSBs in non-cycling mammalian cells (*Figure 2H and I*).

## DNA ends are resected in non-cycling LIN37-deficient cells

The CtIP nuclease is required to initiate DNA end resection during HR and DNA ends in non-cycling abl pre-B cells that are deficient in H2AX undergo resection that depends on CtIP (*Sartori et al., 2007*; *Helmink et al., 2011*). To determine whether IR-induced RPA association with chromatin in LIN37-deficient abl pre-B cells also depends on CtIP, *Lig4$^{-/-}$:Trp53bp1$^{-/-}$*, and *Lig4$^{-/-}$:Lin37$^{-/-}$* abl pre-B cells that expressed a *Ctip* gRNA were treated with doxycycline to induce Cas9 expression and maximal CtIP protein reduction, due to *Ctip* gene inactivation, prior to analysis (*Figure 3A*). This approach, which we refer to as 'bulk gene inactivation', allows for the depletion of proteins normally required for cell division and viability. The chromatin-bound RPA in IR treated non-cycling *Lig4$^{-/-}$: Lin37$^{-/-}$* and *Lig4$^{-/-}$:Trp53bp1$^{-/-}$* abl pre-B cells depended on CtIP (*Figure 3A*), indicating that the increased level of chromatin-bound RPA after IR in cells lacking LIN37 or 53BP1 is due to DNA end resection that would lead to the generation of ssDNA.

CDK-dependent phosphorylation of CtIP is required to promote DNA end resection in cycling cells (*Huertas and Jackson, 2009*; *Huertas et al., 2008*). We treated non-cycling *Lig4$^{-/-}$:Lin37$^{-/-}$* and *Lig4$^{-/-}$:Trp53bp1$^{-/-}$* abl pre-B cells with the CDK4/6 inhibitor Palbociclib at a concentration sufficient to arrest abl pre-B cells in G$_1$ (*Figure 3—figure supplement 1A*) and reduce RB phosphorylation (*Figure 3—figure supplement 1B*), which are both CDK4/6-dependent (*Pennycook and Barr, 2020*; *Weinberg, 1995*). Palbociclib treatment of imatinib-treated non-cycling *Lig4$^{-/-}$:Lin37$^{-/-}$* and *Lig4$^{-/-}$:Trp53bp1$^{-/-}$* abl pre-B cells did not affect the levels of chromatin-bound RPA after IR, indicating CtIP-dependent resection in non-cycling cells does not require CDK4/6 activity (*Figure 3—figure supplement 1C*).

To gain direct evidence of DNA DSB end resection in LIN37-deficient abl pre-B cells, DNA End Sequencing (End-seq) was used to directly assay DNA end structures at approximately 200 *AsiSI* sites in abl pre-B cells with an inducible *AsiSI* endonuclease (*iAsiSI*) (*Canela et al., 2016*). End-seq allows for nucleotide resolution mapping of length and end position of DNA end resection at DSBs generated throughout the genome (*Canela et al., 2016*). The End-seq analysis revealed that while *AsiSI* DSBs in non-cycling *Lig4$^{-/-}$:iAsiSI* abl pre-B cells were minimally resected (<200 bp), those in non-cycling *Lig4$^{-/-}$:Lin37$^{-/-}$:iAsiSI* abl pre-B cells were resected up to 2 kb (*Figure 3B and C*). We conclude that loss of LIN37 leads to the CtIP-dependent resection of broken DNA ends in non-cycling cells.

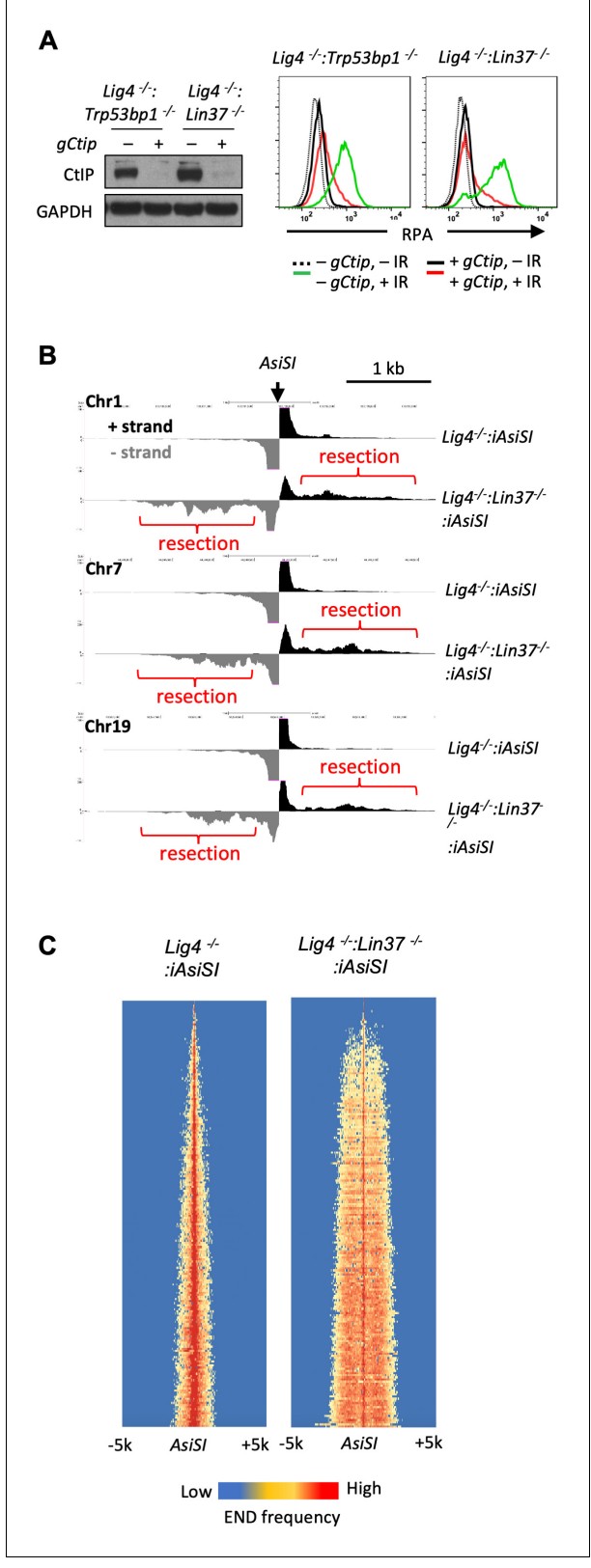

**Figure 3.** LIN37 prevents DNA end resection in non-cycling cells. (**A**) Cas9-induced $Lig4^{-/-}:Trp53bp1^{-/-}$ and $Lig4^{-/-}:Lin37^{-/-}$ abl pre-B cells with (+) and without (−) the $Ctip$ gRNA ($gCtip$). Western blot analysis with indicated antibodies (left) and flow cytometric analysis of chromatin-bound RPA before and after IR of the non-cycling cells (right) are shown. Representative of three experiments. (**B**) End-seq tracks of representative $AsiSI$ sites

*Figure 3 continued on next page*

on mouse chromosomes 1, 7, and 19 in non-cycling *Lig4*$^{-/-}$ and *Lig4*$^{-/-}$:*Lin37*$^{-/-}$ abl pre-B cells. (C) The heatmaps of End-seq at *AsiSI* DSBs across the mouse genome (y-axis) after *AsiSI* induction in non-cycling *Lig4*$^{-/-}$ and *Lig4*$^{-/-}$:*Lin37*$^{-/-}$ abl pre-B cells. Two experiments were carried out in two independently generated *Lig4*$^{-/-}$:*iAsiSI* and *Lig4*$^{-/-}$:*Lin37*$^{-/-}$:*iAsiSI* clones. DSB, double-strand break; End-seq, End Sequencing; gRNA, guide RNA; IR, ionizing radiation.

The online version of this article includes the following figure supplement(s) for figure 3:

**Figure supplement 1.** CtIP promotes resection in non-cycling abl pre-B cells independent of CDK4/6 activity.

## LIN37 and 53BP1 are in distinct pathways of DNA end protection

53BP1 and its downstream effector proteins protect DNA ends from resection through multiple potential mechanisms (**Setiaputra and Durocher, 2019**; **Mirman and de Lange, 2020**; **Bunting et al., 2010**). To determine whether LIN37 functions in the same pathway as 53BP1, we first examined whether loss of LIN37 alters the expression levels of key proteins in the 53BP1 pathway. In this regard, western blot analysis revealed that loss of LIN37 did not lead to reduction in the levels of 53BP1, RIF1, or SHLD1 proteins in cycling or non-cycling abl pre-B cells (**Figure 4A** and **Figure 4—figure supplement 1A**). Moreover, after IR treatment, robust and near equivalent numbers of 53BP1 and RIF1 foci form in non-cycling *Lig4*$^{-/-}$ and *Lig4*$^{-/-}$:*Lin37*$^{-/-}$ abl pre-B cells, demonstrating that

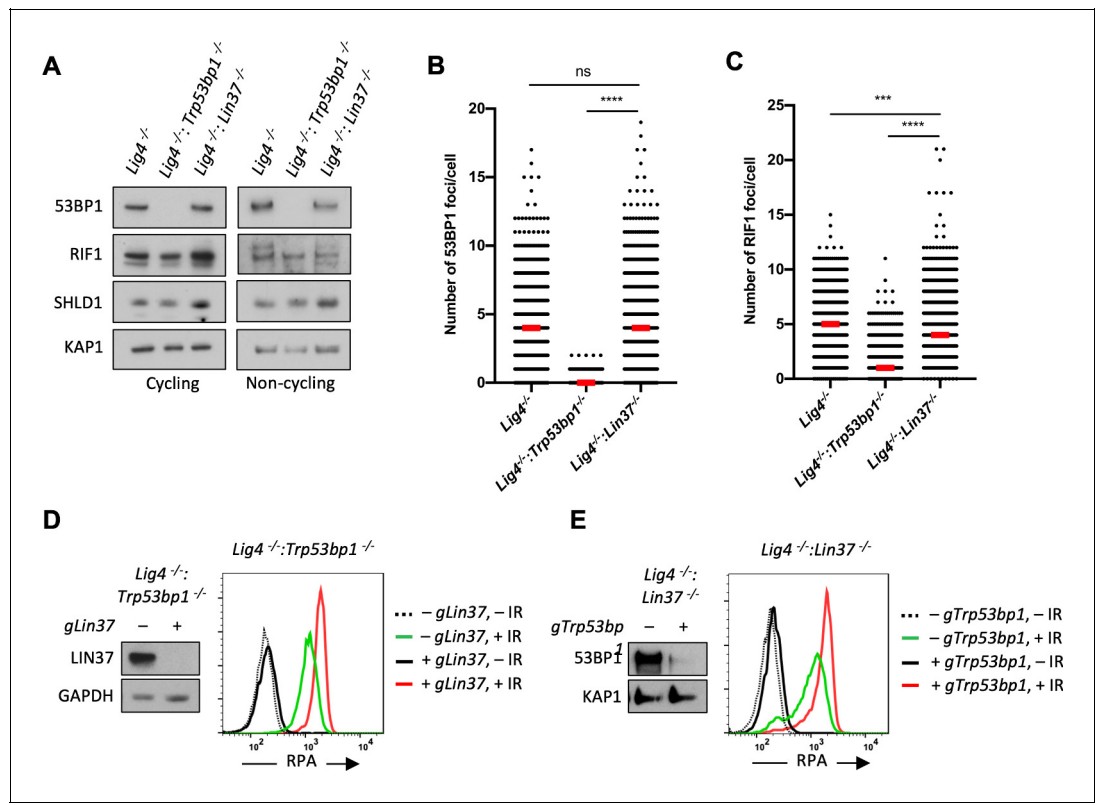

**Figure 4.** 53BP1 and LIN37 have distinct DNA end protection functions. (A) Western blot analysis of indicated proteins in cycling and non-cycling *Lig4*$^{-/-}$, *Lig4*$^{-/-}$:*Trp53bp1*$^{-/-}$, and *Lig4*$^{-/-}$:*Lin37*$^{-/-}$ abl pre-B cells. (B, C) Quantification of 53BP1 (B) or RIF1 (C) foci after IR treatment of non-cycling *Lig4*$^{-/-}$, *Lig4*$^{-/-}$:*Trp53bp1*$^{-/-}$, and *Lig4*$^{-/-}$:*Lin37*$^{-/-}$ abl pre-B cells. Red bars indicate the median number of foci in each sample. More than 1000 cells were analyzed in each cell line in two independent experiments (****p<0.0001, ***p=0.0002, Mann-Whitney test). (D, E) Flow cytometric analysis of chromatin-bound RPA before and after IR of non-cycling *Lig4*$^{-/-}$:*Trp53bp1*$^{-/-}$ (D) or *Lig4*$^{-/-}$:*Lin37*$^{-/-}$ (E) abl pre-B cells after bulk gene inactivation of *Lin37* (gLin37) or *Trp53bp1* (gTrp53bp1), respectively. Representative of three experiments. IR, ionizing radiation.

The online version of this article includes the following figure supplement(s) for figure 4:

**Figure supplement 1.** 53BP1 and LIN37 have distinct DNA end protection functions.

both 53BP1 and RIF1 efficiently localize to DSBs in LIN37-deficient cells (*Figure 4B and C*, *Figure 4—figure supplement 1B and C*).

To test whether 53BP1 and LIN37 protect DNA ends using the same or different pathways, we conducted an epistasis analysis of DNA end resection in non-cycling cells lacking LIN37, 53PB1, or both of these proteins. To this end, a *Lin37* gRNA was used to carry out bulk *Lin37* inactivation in *Lig4$^{-/-}$:Trp53bp1$^{-/-}$* abl pre-B cells (*Figure 4D*). Loss of LIN37 in non-cycling *Lig4$^{-/-}$:Trp53bp1$^{-/-}$* abl pre-B cells led to an increase in chromatin-bound RPA after IR, as compared to non-cycling *Lig4$^{-/-}$:Trp53bp1$^{-/-}$* abl pre-B cells that express LIN37 (*Figure 4D*). Similarly, the loss of 53BP1 in non-cycling *Lig4$^{-/-}$:Lin37$^{-/-}$* abl pre-B cells also led to increased chromatin-bound RPA after IR (*Figure 4E*). Given the additive effect of inactivation of both LIN37 and 53BP1, we conclude that in non-cycling cells 53BP1 and LIN37 are part of mechanistically distinct pathways that are both required to protect DNA ends from resection.

## LIN37 suppresses the expression of DNA end resection and HR proteins in non-cycling cells

As LIN37 participates in the formation of the DREAM transcriptional repressor, we considered the possibility that LIN37 may protect DNA ends through the suppression of genes encoding pro-resection proteins (*Sadasivam and DeCaprio, 2013*). A mutant form of LIN37, LIN37$^{CD}$, does not associate with the DREAM complex and while DREAM lacking LIN37 binds to its target genes, it cannot repress their expression (*Mages et al., 2017*). Expression of WT LIN37, but not LIN37$^{CD}$, in non-cycling *Lig4$^{-/-}$:Lin37$^{-/-}$* abl pre-B cells decreased IR-induced chromatin-bound RPA (*Figure 5A*). That DNA end protection in LIN37-deficient cells can be restored by WT LIN37, but not LIN37$^{CD}$, suggests that LIN37 functions to protect DNA ends through its activity in the DREAM transcriptional repressor.

To identify genes repressed by DREAM that encode proteins that would promote end resection in non-cycling cells, we carried out RNA sequencing (RNA-Seq) analysis of cycling and non-cycling *Lig4$^{-/-}$* and *Lig4$^{-/-}$:Lin37$^{-/-}$* abl pre-B cells. Cycling *Lig4$^{-/-}$:Lin37$^{-/-}$* abl pre-B cells exhibit very few (approximately 20) significant gene expression changes as compared to *Lig4$^{-/-}$* abl pre-B cells (*Figure 5—source data 1*). In contrast, when compared to non-cycling *Lig4$^{-/-}$* abl pre-B cells, non-cycling *Lig4$^{-/-}$:Lin37$^{-/-}$* abl pre-B cells exhibited increased expression (>2-fold) of over 300 genes (*Figure 5B* and *Figure 5—source data 1*). We found that many genes upregulated in non-cycling *Lig4$^{-/-}$:Lin37$^{-/-}$* abl pre-B cells have functions in HR, including DNA end resection (*Brca1*, *Bard1*, *Blm*, and *Exo1*), recombination (*Brca2*, *Bard1*, *Palb2*, *Mms22l*, *Rad51*, *Rad51b*, and *Rad54b*), and DNA synthesis (*Hrob/BC030867*, *Mcm3*, *Mcm4*, *Mcm5*, *Mcm7*, *Mcm8*, *Pold1*, and *Pole*). A subset of FA genes also exhibited increased expression in non-cycling *Lig4$^{-/-}$:Lin37$^{-/-}$* abl pre-B cells (*Figure 5B* and *Figure 5—source data 1*). Many of the upregulated genes we identified were also found to be bound by DREAM in NIH3T3 cells (*Figure 5—source data 1*; *Mages et al., 2017*; *Litovchick et al., 2007*). Gene ontology (GO) analysis reveals that the upregulated genes in non-cycling *Lig4$^{-/-}$:Lin37$^{-/-}$* abl pre-B cells are enriched in biological processes, such as cell cycle, cell division, DNA metabolic process, and DNA repair, consistent with previous analyses in LIN37-deficient mouse cells or human cells deficient in other DREAM components (*Figure 5—figure supplement 1A* and *Figure 5—source data 1*; *Mages et al., 2017*; *Litovchick et al., 2007*). These data indicate that in non-cycling abl pre-B cells LIN37-DREAM transcriptionally represses many genes encoding proteins that function in DNA end resection and HR.

Western blot analysis revealed that the levels of BRCA1, BARD1, BLM, RAD51, and FANCD2 proteins were extremely low in non-cycling *Lig4$^{-/-}$* abl pre-B cells compared to cycling cells (*Figure 5C*). Consistent with their increased transcript levels, the protein levels of BRCA1, BARD1, BLM, RAD51, and FANCD2 were significantly elevated in non-cycling *Lin37$^{-/-}$* and *Lig4$^{-/-}$:Lin37$^{-/-}$* abl pre-B cells as compared to WT and *Lig4$^{-/-}$* abl pre-B cells, respectively (*Figure 5C* and *Figure 5—figure supplement 1B*). In contrast, the levels of these proteins were not altered in cycling LIN37-deficient cells (*Figure 5C* and *Figure 5—figure supplement 1B*). The increased expression of these HR proteins was also observed in non-cycling, but not cycling, *Lin37$^{-/-}$* human MCF10A cells, indicating that LIN37 represses the expression of key DNA end resection and HR proteins to potentially limit DNA end resection in both human and murine non-cycling cells (*Figure 5D*).

The ability of LIN37 to repress the expression of these key HR proteins in non-cycling cells requires LIN37 to function in the context of the DREAM complex as evidenced by our observation

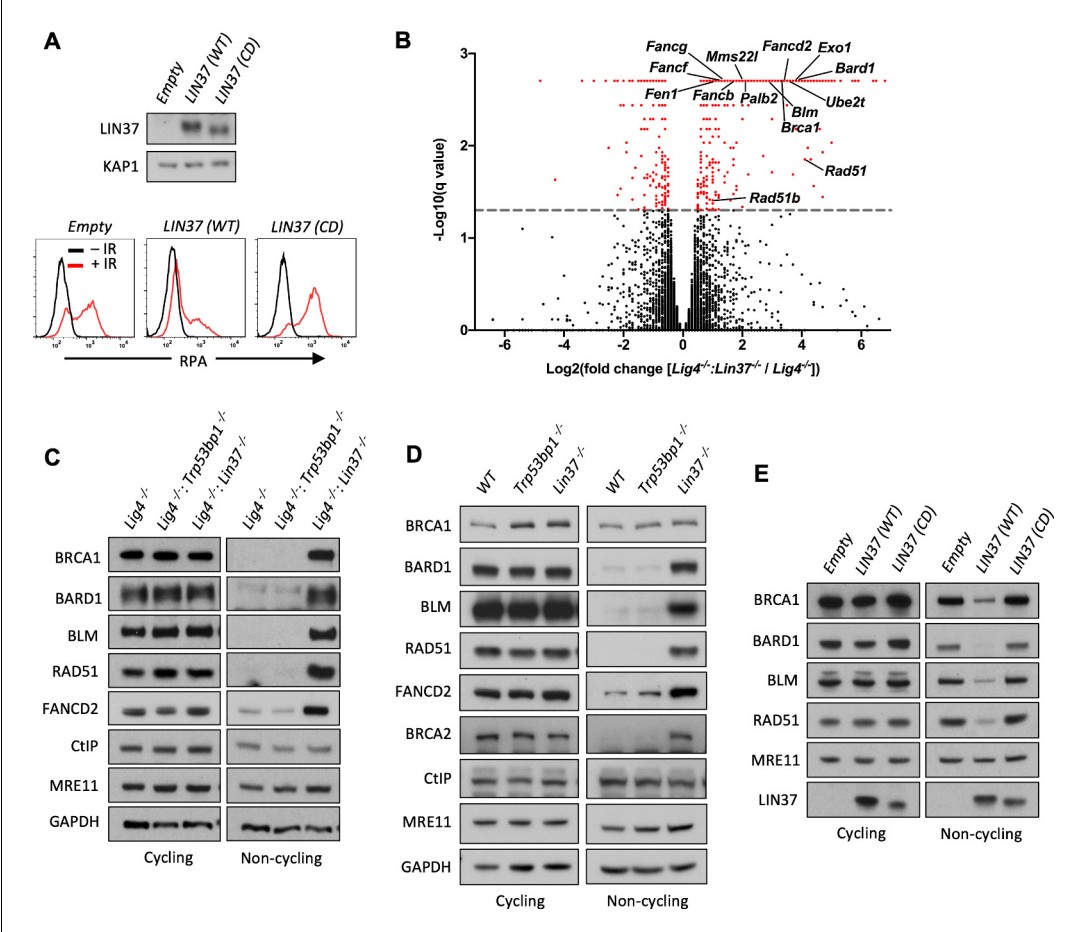

**Figure 5.** LIN37 suppresses the expression of HR protein expression in non-cycling cells. (**A**) Western blot analysis (top) and flow cytometric analysis for chromatin-bound RPA after before or after IR (bottom) of non-cycling $Lig4^{-/-}:Lin37^{-/-}$ abl pre-B cells with empty lentivirus or lentivirus expressing wild type (WT) LIN37 or the LIN37 (CD) mutant. Representative of three experiments. (**B**) Volcano plot of RNA-Seq analysis of non-cycling $Lig4^{-/-}$ and $Lig4^{-/-}:Lin37^{-/-}$ abl pre-B cells showing log2 values of the ratio of normalized transcript levels of $Lig4^{-/-}:Lin37^{-/-}$ to $Lig4^{-/-}$ cells (X-axis) and −log10 of the q-values of fold enrichment of each gene (Y-axis). The dashed line indicates q=0.05. Genes with q-values≤0.05 are denoted as red dots. (**C**) Western blot analysis of indicated proteins in cycling and non-cycling $Lig4^{-/-}$, $Lig4^{-/-}:Trp53bp1^{-/-}$, and $Lig4^{-/-}:Lin37^{-/-}$ abl pre-B cells. (**D**) Western blot analysis of indicated proteins in cycling or non-cycling WT, $Trp53bp1^{-/-}$, and $Lin37^{-/-}$ MCF10A cells. (**E**) Western blot analysis of indicated proteins in cycling and non-cycling $Lig4^{-/-}:Lin37^{-/-}$ abl pre-B cells with empty lentivirus or lentivirus expressing WT LIN37 or the LIN37 (CD) mutant. HR, homologous recombination; IR, ionizing radiation; RNA-Seq, RNA sequencing.

The online version of this article includes the following source data and figure supplement(s) for figure 5:

**Source data 1.** RNA-Seq result and GO analysis in non-cycling $Lig4^{-/-}$ and $Lig4^{-/-}:Lin37^{-/-}$ abl pre-B cells.

**Figure supplement 1.** LIN37 suppresses HR protein expression in non-cycling cells.

that the expression of WT LIN37, but not LIN37$^{CD}$ in non-cycling $Lig4^{-/-}:Lin37^{-/-}$ abl pre-B cells, led to a decrease in the levels of BRCA1, BARD1, BLM, RAD51, and FANCD2 proteins (*Figure 5E*). Of note is that the expression of the MRE11 and CtIP nucleases were not affected by the loss of LIN37 in cycling or non-cycling cells (*Figure 5C* and *Figure 5—figure supplement 1B*). We conclude that in non-cycling cells, LIN37, as part of the DREAM transcriptional repressor complex, represses genes encoding many proteins that function in DNA end resection and DSB repair by HR.

## DNA resection and HR machinery are functional in non-cycling LIN37-deficient cells

We performed a whole-genome CRISPR/Cas9 reverse genetic screen to identify genes that mediate DNA end resection in non-cycling LIN37-deficient abl pre-B cells. To this end, we identified gRNAs enriched in IR treated non-cycling $Lig4^{-/-}:Lin37^{-/-}$ abl pre-B cells that had low levels of chromatin-

bound RPA, indicating reduced DNA end resection despite having no LIN37 (*Figure 6—source data 1*). As expected, gRNAs to *Rbbp8* (encodes CtIP) and to *Mre11b* (encodes MRE11) were enriched in these RPA low cells, in agreement with their nucleolytic roles in resection and emphasizing the validity of our screen (*Figure 6—source data 1*). In addition, we isolated gRNAs to many genes encoding DNA end resection and HR proteins that are normally repressed by LIN37, including *Brca1*, *Bard1*, *Blm*, and several FA genes, indicating that the expression of these proteins brought about by LIN37-deficiency in non-cycling cells promotes the extensive DNA end resection that is observed in these cells (*Figure 6—source data 1*). Indeed, bulk inactivation of BRCA1, BARD1, BLM, or FANCD2 significantly reduced the level of chromatin-bound RPA after IR in non-cycling $Lig4^{-/-}$:$Lin37^{-/-}$ abl

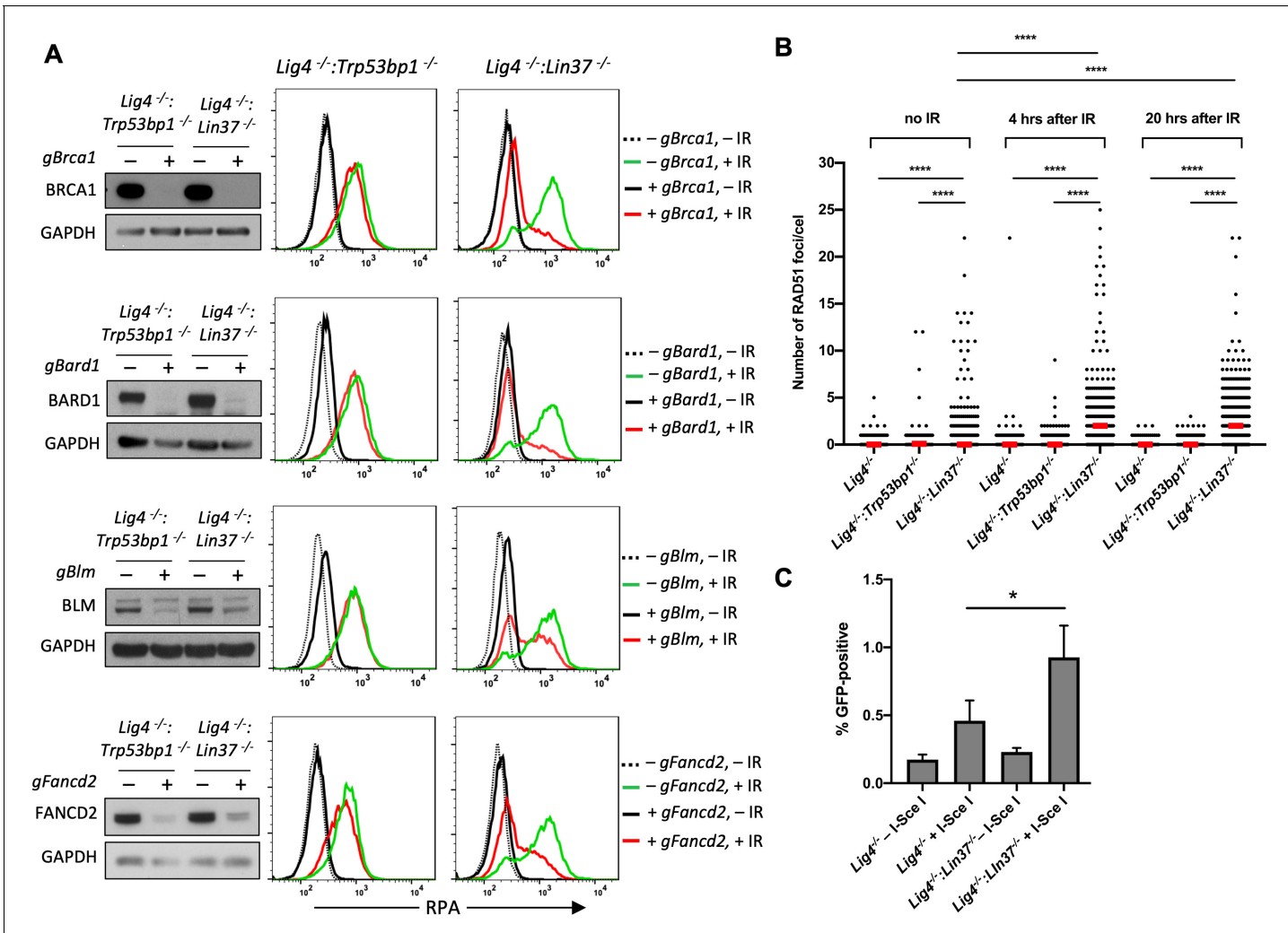

**Figure 6.** LIN37 prevents resection and HR through suppressing HR protein expression in non-cycling cells. (A) Western blot analysis of proliferating $Lig4^{-/-}$:$Trp53bp1^{-/-}$ or $Lig4^{-/-}$:$Lin37^{-/-}$ abl pre-B cells with or without indicated gRNAs following Cas9 induction for bulk gene inactivation using the indicated antibodies (left). Flow cytometric analysis of chromatin-bound RPA before and after IR of the same cells after rendered non-cycling by imatinib treatment (right). Representative of three experiments. (B) Quantification of RAD51 foci in non-cycling $Lig4^{-/-}$, $Lig4^{-/-}$:$Trp53bp1^{-/-}$, and $Lig4^{-/-}$:$Lin37^{-/-}$ abl pre-B cells before IR treatment and 4 and 20 hr after IR. Red bars indicate the median number of RAD51 foci in each sample of more than 1000 cells analyzed for each cell line. Representative of two independent experiments (****p<0.0001, Mann-Whitney test). (C) Flow cytometric analysis of HR-mediated DSB repair in non-cycling $Lig4^{-/-}$ and $Lig4^{-/-}$:$Lin37^{-/-}$ abl pre-B cells using the HPRT-DR-GFP reporter. The percentage of GFP-positive cells is shown. Error bars are ± SEM from three experiments (*p=0.0124, t-test). DSB, double-strand break; gRNA, guide RNA; HR, homologous recombination; IR, ionizing radiation.

The online version of this article includes the following source data and figure supplement(s) for figure 6:

**Source data 1.** RPA screen result in non-cycling $Lig4^{-/-}$:$Lin37^{-/-}$ abl pre-B cells.

**Figure supplement 1.** LIN37 deficiency leads to RAD51 focus formation in non-cycling abl pre-B cells.

pre-B cells (*Figure 6A*). In striking contrast, the loss of these proteins had no effect on RPA localization to DSBs after IR treatment of non-cycling *Lig4⁻/⁻:Trp53bp1⁻/⁻* abl pre-B cells, further emphasizing that LIN37 and 53BP1 are part of mechanistically distinct pathways that prevent DNA end resection and ssDNA generation in non-cycling cells (*Figure 6A*).

LIN37-deficient non-cycling abl pre-B cells also exhibited induction of *Brca2*, *Palb2*, and *Rad51*, which function to replace RPA with RAD51 on ssDNA to form RAD51 nucleofilaments at DSBs during HR (*Figure 5B, C and D*, *Figure 5—figure supplement 1B* and *Figure 5—source data 1*). Indeed, there was a significant increase in the number of RAD51 foci in non-cycling *Lig4⁻/⁻:Lin37⁻/⁻* abl-pre-B cells after IR as compared to *Lig4⁻/⁻* abl-pre-B cells (*Figure 6B* and *Figure 6—figure supplement 1*). Once formed, a RAD51 nucleofilament will initiate a homology search and HR-mediated DSB repair (*Prakash et al., 2015*). Indeed, using the HPRT-DR-GFP reporter for DSB repair by HR, we observed an increase in homology-mediated DSB repair in non-cycling *Lig4⁻/⁻:Lin37⁻/⁻* abl-pre-B cells as compared to *Lig4⁻/⁻* abl-pre-B cells (*Figure 6C*; *Pierce et al., 2001*).

We conclude that in the absence of LIN37, the extensive DNA end resection can lead to HR-mediated DSB repair in non-cycling cells. Moreover, this occurs in the presence of 53BP1.

## LIN37 uniquely prevents DNA end resection in quiescent $G_0$ cells

We find that LIN37 is required to prevent DNA end resection and subsequent HR steps in non-cycling cells that have 2N DNA content and do not incorporate BrdU indicating that they are in $G_0$ or $G_1$. To distinguish between $G_0$ and $G_1$, we assayed non-cycling imatinib-treated abl pre-B cells and EGF-deprived MCF10A cells for expression of the cyclin-dependent kinase, CDK4, and phospho-CDK4, which along with CDK6 is required for non-cycling quiescent $G_0$ cells to move into $G_1$ (*Figure 7A and B*; *Malumbres and Barbacid, 2001*; *Pennycook and Barr, 2020*). Both non-cycling abl pre-B cells and MCF10A cells had low levels of CDK4 and phspho-CDK4 indicative of them being in $G_0$ (*Figure 7A and B*). RB suppresses genes encoding proteins required for $G_0$ cells to transit to $G_1$ and then into S-phase (*Weinberg, 1995*; *Pennycook and Barr, 2020*; *Sadasivam and DeCaprio, 2013*). The phosphorylation of RB by CDK4 or CDK6 leads to its inactivation and the transit of cells from $G_0$ to $G_1$ (*Weinberg, 1995*; *Pennycook and Barr, 2020*). Phospho-Rb was not detected by western blot analysis of imatinib-treated abl pre-B cells or MCF10A cells deprived of EGF indicating that these cells are in $G_0$ (*Figure 7A and B*). PCNA, which is expressed early in the transition from $G_1$ to S was also not detected in these cells (*Figure 7A and B*). Taken together, these data indicate that imatinib-treated abl pre-B cells and MCF10A cells deprived of EGF have exited the cell cycle and entered $G_0$, also known as quiescence, and LIN37 is required to protect DNA ends from resection in these cells.

Cycling cells in $G_1$-phase also rely on NHEJ to repair DNA DSBs and we next asked whether LIN37 functions to protect DNA ends from resection in these cells. To this end, we employed the PIP-FUCCI cell cycle sensor to isolate $G_1$ and S/$G_2$/M populations from cycling *Lig4⁻/⁻*, *Lig4⁻/⁻:Trp53bp1⁻/⁻*, and *Lig4⁻/⁻:Lin37⁻/⁻* abl pre-B cells and WT, *Trp53bp1⁻/⁻*, and *Lin37⁻/⁻* MCF10A cells by flow cytometric cell sorting (*Grant et al., 2018*). The effectiveness of this purification was evidenced by the absence of Cyclin A in $G_1$-phase cells (*Figure 7C and D*; *Henglein et al., 1994*). We conducted RNA-Seq analysis on proliferating $G_1$-phase *Lig4⁻/⁻* and *Lig4⁻/⁻:Lin37⁻/⁻* abl pre-B cells to determine if the LIN37-DREAM complex also exerts its transcription repressor activity in the $G_1$-phase of the cell cycle. Our RNA-Seq analysis revealed that the expression of ~360 genes is significantly elevated in $G_1$ *Lig4⁻/⁻:Lin37⁻/⁻*, compared to *Lig4⁻/⁻* abl pre-B cells (*Figure 7—source data 1*). Of these genes, ~200 are uniquely upregulated in proliferating $G_1$-phase *Lig4⁻/⁻:Lin37⁻/⁻* abl pre-B cells while the remaining genes have increased expression in both imatinib-treated $G_0$ and proliferating $G_1$-phase *Lig4⁻/⁻:Lin37⁻/⁻* abl pre-B cells, including *Brca1*, *Bard1*, *Blm*, *Rad51*, and *Fancd2* (*Figure 7—source data 2*). However, western blot analysis revealed that in contrast to $G_0$ cells, $G_1$ cells expressed readily detectable levels of the HR proteins BRCA1, BARD1, BLM, FANCD2, and RAD51. Moreover, loss of LIN37 did not lead to a significant change in the levels of these proteins in $G_1$-phase cells isolated from proliferating populations (*Figure 7C and D*). These results indicate that LIN37-DREAM functions to negatively regulate the expression of a subset of HR genes in $G_1$-phase cells, however, this does not lead to a significant decrease in the levels of proteins encoded by these genes.

We next asked whether LIN37 functions to protect DNA ends from extensive end resection in cycling $G_1$-phase cells. To do this, we incubated proliferating *Lig4⁻/⁻*, *Lig4⁻/⁻:Trp53bp1⁻/⁻*, and

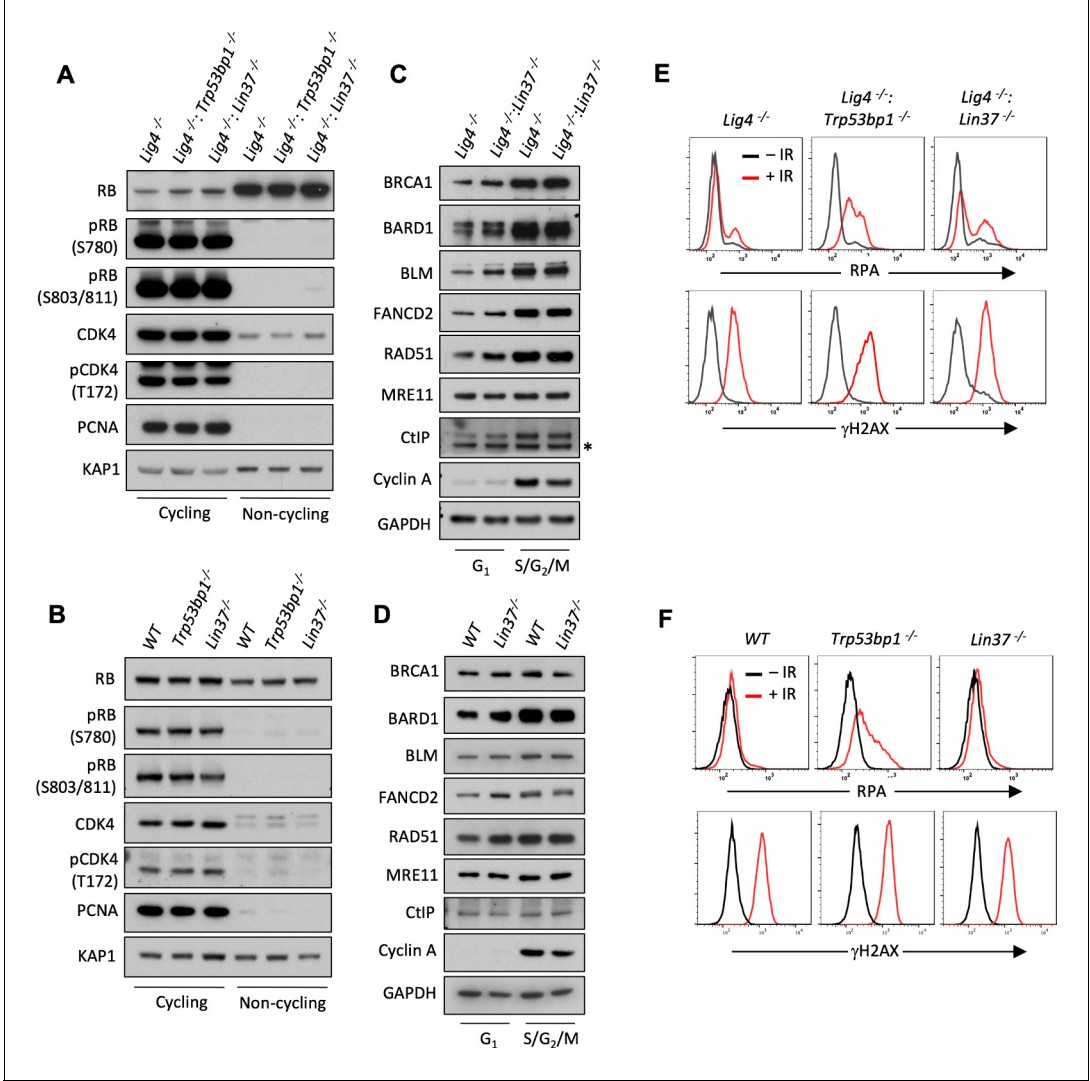

**Figure 7.** LIN37 function in DNA end protection is restricted to $G_0$. (**A, B**) Western blot analysis of indicated proteins in cycling and non-cycling abl pre-B cells (**A**) or MCF10A cells (**B**). (**C, D**) Western blot analysis of indicated proteins in cycling $G_1$ or $S/G_2/M$ abl pre-B cells (**C**) or MCF10A cells (**D**), isolated by flow cytometric cell sorting based on the PIP-FUCCI reporter. Representative of two independent experiments. Asterisk indicates non-specific recognizing bands. (**E, F**) Flow cytometric analysis of chromatin-bound RPA and $\gamma$H2AX before and after IR treatment of $G_1$-phase $Lig4^{-/-}$, $Lig4^{-/-}:Trp53bp1^{-/-}$ and $Lig4^{-/-}:Lin37^{-/-}$ abl pre-B cells (**E**) or WT, $Trp53bp1^{-/-}$, and $Lin37^{-/-}$ MCF10A cells (**F**). Representative of three experiments. IR, ionizing radiation; WT, wild type .

The online version of this article includes the following source data and figure supplement(s) for figure 7:

**Source data 1.** RNA-Seq result and GO analysis in cycling $G_1$ $Lig4^{-/-}$ and $Lig4^{-/-}:Lin37^{-/-}$ abl pre-B cells.

**Source data 2.** GO analysis of genes upregulated in non-cycling $G_0$ and/or cycling $G_1$ $Lig4^{-/-}:Lin37^{-/-}$ abl pre-B cells.

**Figure supplement 1.** Identification of $G_1$-phase cells in proliferating cells.

$Lig4^{-/-}:Lin37^{-/-}$ abl pre-B cells and WT, $Trp53bp1^{-/-}$, and $Lin37^{-/-}$ MCF10A cells in media containing EdU, which was incorporated into newly synthesized DNA allowing for the detection of S-phase cells. After IR, RPA association with DNA DSBs was assayed by flow cytometry in $G_1$-phase cells, which were identified as cells being EdU-negative and having 2N DNA content (**Figure 7—figure supplement 1A and B**). In contrast to $G_0$ cells, which are dependent on 53BP1 and LIN37 for DNA end protection, $G_1$ cells appear to depend primarily on 53BP1 as evidenced by the substantial increase in RPA association with chromatin in IR treated 53BP1-deficient $G_1$-phase cells as compared to LIN37-deficient $G_1$-phase cells (**Figure 7E and F**). We conclude that LIN37 functions to prevent

DNA end resection primarily in quiescent $G_0$ cells while 53BP1 functions in both proliferating $G_1$ and quiescent $G_0$ phase cells.

## Discussion

Rb and the DREAM complex are transcriptional repressors that silence the expression of genes required to promote cell cycle, driving cells to exit the cell cycle and enter $G_0$ or quiescence, where most cells in the human body reside (*Mages et al., 2017*; *Weinberg, 1995*). Quiescent $G_0$ cells and cycling cells in $G_1$-phase have 2N DNA content and rely on NHEJ to repair DNA DSBs and maintain genome stability. 53BP1 and its downstream effectors are required to promote NHEJ in $G_0$ and $G_1$ cells by antagonizing DNA end resection and ssDNA generation (*Mirman and de Lange, 2020*; *Setiaputra and Durocher, 2019*). However, here, we show that in $G_0$ cells, DNA end protection also requires LIN37, a component of the DREAM complex. Moreover, LIN37 function also prevents aberrant homology-mediated joining in $G_0$ cells.

LIN37 association with DREAM is required for its transcriptional repressor function, but is not required for the binding of the complex to its target genes (*Mages et al., 2017*). In addition to genes encoding proteins that promote cell cycle, DREAM complex binding sequences, termed cell cycle genes homology regions (CHRs), are also found in the promoter regions of many HR genes (*Mages et al., 2017*; *Müller et al., 2014*). Several lines of evidence presented here support the notion that in $G_0$ cells LIN37 functions through DREAM transcriptional repression to protect DNA ends from resection. Loss of LIN37 in $G_0$ cells leads to the expression of many HR genes that promote DNA end resection such as *Brca1*, *Bard1*, *Blm*, and *Fancd2* (*Figure 5B* and *Figure 5—source data 1*). Moreover, this increased gene expression leads to a significant increase in the levels of these proteins in $G_0$ cells (*Figure 5C and D* and *Figure 5—figure supplement 1B*). Loss of BRCA1, BARD1, BLM, or FANCD2 in LIN37-deficient $G_0$ cells prevents DNA end resection demonstrating that the function of each of these proteins is required to promote DNA end resection in these cells (*Figure 6A*). Finally, while the expression of WT LIN37 in LIN37-deficient $G_0$ cells prevents the expression of HR genes and DNA end resection the expression of LIN37$^{CD}$, which cannot participate in forming a functional DREAM repressor complex, does not (*Figure 5E*).

Why is it that LIN37 is the only subunit of the DREAM complex that was identified in our screen? One potential explanation is the functional redundancy of some DREAM subunits. For example, it has been suggested that Rb-like protein RBL1/p107 and RBL2/p130 can both function in the DREAM complex and therefore inactivation of RBL1 or RBL2 may not significantly impact the activity of the DREAM complex (*Litovchick et al., 2007*). The same consideration applies to the inhibitory E2F subunits E2F4 and E2F5. Additionally, several components of the DREAM complex are required to form the MuvB subcomplex which functions outside of $G_1/G_0$-phase to promote the expression of genes required for essential processes such as DNA replication in S or $G_2$ phase cells (e.g., the DNA binding component LIN54 *Marceau et al., 2016*; *Schmit et al., 2009*). Inactivating these subunits may impact cell proliferation in a way that does not allow them to be present or selected in our screen.

LIN37-DREAM and 53BP1 function in distinct ways to protect DNA ends from aberrant resection. This is supported by the additive effect of 53BP1 and LIN37 deficiency on DNA end resection in *Lig4$^{-/-}$* $G_0$ abl pre-B cells (*Figure 4D and E*). Resection in $G_0$ LIN37-deficient cells relies on BRCA1, BARD1, BLM, and FANCD2 as evidenced by the lack of DNA end resection in LIN37-deficient cells that have lost expression of any of these proteins (*Figure 6A*). In contrast, 53BP1-deficient $G_0$ cells express low levels of BRCA1, BARD1, BLM, and FANCD2 and loss of expression of these proteins does not impact DNA end resection in these cells (*Figure 6A*). Thus, in $G_0$ cells LIN37 prevents BRCA1-dependent DNA end resection whereas 53BP1 protects DNA ends from BRCA1-independent resection pathways.

Cycling cells in $G_1$-phase also must repair DSBs by NHEJ and thus need to protect DNA ends from excess resection. In contrast to what we observe in $G_0$ cells, the loss of LIN37 in cycling cells in $G_1$-phase does not lead to alterations in HR protein levels or aberrant resection of DNA ends as evidenced by increased RPA association at DSBs (*Figure 7*). In $G_1$-phase cells, 53BP1-RIF1 prevents BRCA1-CtIP from associating with DSBs and promoting resection (*Escribano-Díaz et al., 2013*; *Chapman et al., 2013*). In contrast, in S/$G_2$-phase cells, this regulatory balance is tipped with BRCA1 inhibiting RIF1 association with DSBs, which would otherwise antagonize resection (*Escribano-Díaz et al., 2013*; *Chapman et al., 2013*). Like $G_1$-phase cells, in $G_0$ cells 53BP1 functions to

protect DNA ends from resection. Upon the loss of LIN37 in $G_0$ cells, the expression of HR proteins, including BRCA1, does not inactivate the 53BP1 pathway. Indeed, we find robust 53BP1-RIF1 association with DSBs in LIN37-deficient $G_0$ cells and the loss of 53BP1 in these cells leads to increased RPA association at DSBs indicating that 53BP1 functions in DNA end protection in these cells (*Figure 4B–E*, *Figure 4—figure supplement 1B and C*). We speculate that the genetic program activated by the loss of LIN37 in $G_0$ cells leads to the activation of pathways that regulate the balance of anti-resection 53BP1 activities and pro-resection BRCA1 activities in a manner that favors DNA end resection. Although the identity of these pathways and the manner in which they function is unknown, we show that they do not prevent 53BP1-RIF1 association at DSBs in $G_0$ cells (*Figure 4B and C*).

While LIN37-DREAM and 53BP1 both inhibit DNA end resection in $G_0$ cells, LIN37-DREAM has additional activities in promoting genome stability by preventing resected DNA ends from progressing in HR-mediated DSB repair. LIN37-deficient $G_0$ cells express BRCA2, PALB2, and RAD51 that convert RPA-coated ssDNA at broken DNA ends to RAD51 nucleofilaments that can mediate homology-mediated repair (*Figures 5B–D*, *6B and C*, *Figure 4—figure supplement 1*, *Figure 5—figure supplement 1* and *Figure 3—figure supplement 1*), which in $G_0$ cells could lead to aberrant homology-mediated DNA end joining. We speculate that loss of LIN37-DREAM would not lead to a general defect in NHEJ-mediated DSB repair in $G_0$ cells, which would be expected to occur rapidly before the broken DNA ends can be resected. Rather, LIN37-DREAM function would prevent rare DSBs that are not rapidly repaired from being resected preventing normal NHEJ and promoting aberrant homology-mediated joining that occur in the setting of activating HR in the absence of a homologous sister chromatid. In this case, the homology would be found on the same or another chromosome leading to chromosomal deletions, inversions, and translocations, common types of chromosomal rearrangements. Additionally, if resected DSBs in quiescent cells are repaired through 'allelic recombination' using homologous chromosomes as the repair templates during HR, this could result in loss of heterozygosity (LOH) of the chromosomes undergoing repair. Depending on the homologous chromosomes serving as the repair templates, LOH can also cause loss of chromosomal region in the repaired chromosomes (LOH with copy number loss) or results in the replacement of WT, functional copy of genes with mutated, pathological genes on the homologous chromosomes (LOH with neutral copy number) (*Bielas et al., 2007*). Thus, in quiescent cells, LIN37-DREAM promotes genome stability by both antagonizing DNA end resection and preventing the aberrantly homology-mediated joining of resected DNA ends.

# Materials and methods

**Key resources table**

| Reagent type (species) or resource | Designation | Source or reference | Identifiers | Additional information |
|---|---|---|---|---|
| Antibody | Anti-53BP1 (Rabbit polyclonal) | Bethyl Laboratories | A300-272A | WB (1:3000) |
| Antibody | Anti-53BP1 (Rabbit polyclonal) | Novus Biologicals | NB100-305 | IF (1:1000) |
| Antibody | Anti-LIN37 (Mouse monoclonal) | Santa Cruz Biotechnology | sc-515686 | WB (1:200) |
| Antibody | Anti-BLM (Rabbit polyclonal) | Bethyl Laboratories | A300-572A | WB (1:2000) |
| Antibody | Anti-BRCA1 (Mouse monoclonal) | R and D Systems | Custom made (Andre Nussenzweig, NCI) | WB (1:1000); for mouse BRCA1 |
| Antibody | Anti-BRCA1 (Mouse monoclonal) | Millipore Sigma | 07-434 | WB (1:1000); for human BRCA1 |
| Antibody | Anti-RAD51 (Rabbit polyclonal) | Millipore Sigma | ABE257 | WB (1:2000) |

*Continued on next page*

Continued

| Reagent type (species) or resource | Designation | Source or reference | Identifiers | Additional information |
|---|---|---|---|---|
| Antibody | Anti-RAD51 (Rabbit polyclonal) | Abcam | ab176458 | IF (1:250) |
| Antibody | Anti-BARD1 (Rabbit polyclonal) | Thermo Fisher Scientific | PA5-85707 | WB (1:1000) |
| Antibody | Anti-CtIP (Rabbit polyclonal) | N/A | Custom made (Richard Baer, Columbia University) | WB (1:1000) |
| Antibody | Anti-MRE11 (Rabbit polyclonal) | Novus Biologicals | NB100-142 | WB (1:2000) |
| Antibody | Anti-RIF1 (Rabbit polyclonal) | Abcam | ab13422 | WB (1:500) |
| Antibody | Anti-RIF1 (Rabbit polyclonal) | N/A | Custom made (Davide Robbiani, Rockefeller University) | IF (1:5000) |
| Antibody | Anti-C20orf196/ SHLD1 (Rabbit polyclonal) | Thermo Fisher Scientific | PA5-559280 | WB (1:200) |
| Antibody | Anti-GAPDH (Mouse Monoclonal) | Millipore Sigma | G8795 | WB (1:10,000) |
| Antibody | Anti-KAP1 (Rabbit polyclonal) | Genetex | GTX102226 | WB (1:2000) |
| Antibody | Anti-FANCD2 (Rabbit monoclonal) | R and D Systems | MAB93691 | WB (1:1000) |
| Antibody | Anti-BRCA2 (Rabbit polyclonal) | Proteintech | 19791-1-AP | WB (1:500); for human BRCA2 |
| Antibody | Anti-Rb1 (Mouse monoclonal) | Thermo Fisher Scientific | LF-MA0173 | WB (1:1000) |
| Antibody | Anti-Phospho -Rb (Ser780) (Rabbit polyclonal) | Cell Signaling Technology | 8180T | WB (1:1000) |
| Antibody | Anti-Phospho -Rb (Ser807/ 811) (Rabbit polyclonal) | Cell Signaling Technology | 8516T | WB (1:1000) |
| Antibody | Anti-PCNA (Rabbit polyclonal) | Bethyl Laboratories | A300-276A | WB (1:3000) |
| Antibody | Anti-CDK4 (Rabbit polyclonal) | Novus Biologicals | NBP1-31308 | WB (1:1000) |
| Antibody | Anti-CDK4 (phosphor Thr172) (Rabbit polyclonal) | GeneTex | GTX00778 | WB (1:1000) |
| Antibody | Anti-RPA32 (4E4) (Rat monoclonal) | Cell Signaling Technology | 2208S | WB (1:1000); FC (1:500) |
| Antibody | Anti-phospho- H2AX (ser139) (Mouse monoclonal) | Millipore Sigma | 05-636 | FC (1:1000) |
| Antibody | HRP, goat anti-mouse | Promega | W4021 | WB (1:5000) |

Continued

| Reagent type (species) or resource | Designation | Source or reference | Identifiers | Additional information |
|---|---|---|---|---|
| Antibody | HRP, goat anti-rabbit IgG | Promega | W4011 | WB (1:5000) |
| Antibody | Alexa Fluor 555, donkey anti-rabbit IgG | Thermo Fisher Scientific | A-31572 | IF (1:5000) |
| Antibody | Alexa Fluor 488, goat anti-rat IgG | BioLegend | 405418 | FC (1:500) |
| Antibody | Alexa Fluor 647, goat anti-mouse IgG | BioLegend | 405322 | FC (1:500) |
| Recombinant DNA reagent | pCW-Cas9 (plasmid) | Addgene | 50661 | |
| Recombinant DNA reagent | pX330-U6-Chimeric_BB-CBh-hSpCas9 (plasmid) | Addgene | 42230 | |
| Recombinant DNA reagent | pKLV-U6 gRNA(BbsI)-PGKpuro-2ABFP (plasmid) | Addgene | 50946 | |
| Recombinant DNA reagent | pLenti-CMV-Blast-PIP-FUCCI (plasmid) | Addgene | 138715 | |
| Recombinant DNA reagent | Genome-wide CRISPR guide RNA library V2 (plasmid) | Addgene | 67988 | |
| Recombinant DNA reagent | Lin37 cDNA BC013546 (plasmid) | transOMIC | TCM1004 | |
| Recombinant DNA reagent | TRE-Thy1.1 (plasmid) | This study | N/A | Available upon request |
| Recombinant DNA reagent | pHPRT-DR-GFP (plasmid) | Marian Jasin, MSKCC | N/A | |
| Recombinant DNA reagent | pCBASceI (plasmid) | Marian Jasin, MSKCC | N/A | |
| Recombinant DNA reagent | pCBA (plasmid) | Marian Jasin, MSKCC | N/A | |
| Cell line (*Homo-sapiens*) | *MCA10A* | ATCC | CRL-10317 | |
| Cell line (*Homo-sapiens*) | *MCA10A: iCas9* | This study | Clone 25 | Available upon request |
| Cell line (*Homo-sapiens*) | *MCA10A: Trp53bp1$^{-/-}$: iCas9* | This study | Clones 7 and 50 | Available upon request |
| Cell line (*Homo-sapiens*) | *MCA10A: Lin37$^{-/-}$:iCas9* | This study | Clones 5 and 21 | Available upon request |
| Cell line (*Mus musculus*) | WT:*iCas9* abl pre-B cells | This study | M63.1.MG36.iCas9.302 | Available upon request |
| Cell line (*M. musculus*) | *Trp53bp1$^{-/-}$:iCas9* abl pre-B cells | This study | Clones 1 and 27 | Available upon request |
| Cell line (*M. musculus*) | *Lin37$^{-/-}$:iCas9* abl pre-B cells | This study | Clones 9 and 59 | Available upon request |
| Cell line (*M. musculus*) | *Lig4$^{-/-}$:iCas9* abl pre-B cells | This study | A5.83.MG9.iCas9.16 | Available upon request |

*Continued on next page*

*Continued*

| Reagent type (species) or resource | Designation | Source or reference | Identifiers | Additional information |
|---|---|---|---|---|
| Cell line (*M. musculus*) | *Lig4*−/−*: Trp53bp1*−/−*:iCas9* abl pre-B cells | This study | Clones 81 and 82 | Available upon request |
| Cell line (*M. musculus*) | *Lig4*−/−*:Lin37*−/−*:iCas9* abl pre-B cells | This study | Clones 6 and 42 | Available upon request |
| Chemical compound, drug | Imatinib | Selleckchem | S2475 | |
| Chemical compound, drug | Doxycycline | Sigma-Aldrich | D9891 | |
| Chemical compound, drug | Puromycin | Sigma-Aldrich | P9620 | |
| Chemical compound, drug | EGF | PeproTech | AF-100-15 | |
| Chemical compound, drug | Hydrocortisone | Sigma-Aldrich | H-0888 | |
| Chemical compound, drug | Cholera Toxin | Sigma-Aldrich | C-8052 | |
| Chemical compound, drug | Insulin | Sigma-Aldrich | I-1882 | |
| Commercial assay or kit | Cytofix/ Cytoperm solution | BD Biosciences | 554722 | |
| Commercial assay or kit | Perm/Wash Buffer | BD Biosciences | 554723 | |
| Commercial assay or kit | Click-iT EdU Alexa Fluor 647 Flow Cytometry Assay Kit | Life Technologies | C10419 | |
| Commercial assay or kit | SG Cell Line 4D X Kit L | Lonza | V4XC-3024 | |
| Other | 7-AAD (DNA stain) | BD Biosciences | 559925 | |
| Sequence-based reagent | pKLV lib330F | This study (designed based on *Tzelepis et al., 2016*) | PCR primers | AATGGACTATCA TATGCTTACCGT |
| Sequence-based reagent | pKLV lib490R | This study (designed based on *Tzelepis et al., 2016*) | PCR primers | CCTACCGGTG GATGTGGAATG |
| Sequence-based reagent | PE.P5_pKLV lib195 Fwd | This study (designed based on *Tzelepis et al., 2016* and standard Illumana adaptor sequences) | PCR primers | AATGATACGGC GACCACCG AGATCTGGC TTTATATA TCTTGTGGAAAGGAC |
| Sequence-based reagent | P7 index180 Rev | This study (designed based on *Tzelepis et al., 2016* and standard Illumana adaptor sequences) | PCR primers | CAAGCAGAAGACGG CATACGAGAT *INDEX*GTGACTGG AGTTCAGACGTGTGC TCTTCCGATCCAGA CTGCCTTGGGAAAAGC |
| Sequence-based reagent | Lin37 iso1_5′XhoI_S | This study (designed based on cDNA BC013546) | PCR primers | GCCCTCGAGATGTT CCCGGTAAAGGTG AAAGTGG |

*Continued on next page*

*Continued*

| Reagent type (species) or resource | Designation | Source or reference | Identifiers | Additional information |
|---|---|---|---|---|
| Sequence-based reagent | Lin37 3'NotI_AS | This study (designed based on cDNA BC013546) | PCR primers | GCCGCGGCCGCTCACTGCCGGTCATACATCTCCCGT |
| Sequence-based reagent | Lin37 CD1_AS | This study (designed based on cDNA BC013546 and *Mages et al., 2017*) | PCR primers | TACAGTGGTGTGTTCTCACTGAACTGGGCCAAGTCCACAGCCCCG GCAAATAG CTTGATC |
| Sequence-based reagent | Lin37 CD2_S | This study (designed based on cDNA BC013546 and *Mages et al., 2017*) | PCR primers | ACTTGGCCCAGTTCAGTGAGAACACACCACTGTACCCATCGCCGGCGCCTGGATGCGCA |
| Sequence-based reagent | BU1 | *Canela et al., 2016* | PCR primers | 5'-Phos-GATCGGAAGAGCGTCGTGTAGGGAAAGAGTGUU[Biotin-dT]U[Biotin-dT]UUACACTCTTTC CCTACACGACGCTCTTCCGATC*T-3' [*phosphorothioate bond] |
| Sequence-based reagent | BU2 | *Canela et al., 2016* | PCR primers | 5'-Phos-GATCGGAAGAGCACACG TCUUUUUUUUAGACGTGTGCTCTTCCGATC*T-3' [*phosphorothioate bond] |
| Sequence-based reagent | *53* bp1 gRNA sequence | Sequence is from *Tzelepis et al., 2016* | N/A | GAACCTGTCAGACCCGATC |
| Sequence-based reagent | *Lin37* gRNA sequences | Sequence is from *Tzelepis et al., 2016* | N/A | AAGCTATTTGACCGGAGTG |
| Sequence-based reagent | *Brca1* gRNA sequence | Sequence is from *Tzelepis et al., 2016* | N/A | GTCTACATTGAACTAGGTA |
| Sequence-based reagent | *Ctip* gRNA sequence | Sequence is from *Tzelepis et al., 2016* | N/A | ATTAACCGGCTACGAAAGA |
| Sequence-based reagent | *Bard1* gRNA sequence | Sequence is from *Tzelepis et al., 2016* | N/A | AAATCGTAAAGGCTGCCAC |
| Sequence-based reagent | *Blm* gRNA sequence | Sequence is from *Tzelepis et al., 2016* | N/A | GATTTAACGAAGGAATCGG |
| Sequence-based reagent | *Fancd2* gRNA sequence | Sequence is from *Tzelepis et al., 2016* | N/A | TCTTGTGATGTCGCTCGAC |
| Sequence-based reagent | *Trp53bp1* (human) gRNA sequence | Sequence is from *Tzelepis et al., 2016* | N/A | TCTAGTGTGTTAGATCAGG |

*Continued on next page*

*Continued*

| Reagent type (species) or resource | Designation | Source or reference | Identifiers | Additional information |
|---|---|---|---|---|
| Sequence-based reagent | *Lin37* (human) gRNA sequence | Sequence is from *Tzelepis et al., 2016* | N/A | TCTAGGGAGC GTCTGGATG |
| Software, algorithm | Image J | NIH | RRID:SCR_003070 | |
| Software, algorithm | FlowJo | FlowJo | RRID:SCR_008520 | |
| Software, algorithm | Prism | GraphPad | RRID:SCR_002798 | |
| Software, algorithm | SeqKit | *Shen et al., 2016* | RRID:SCR_018926 | |
| Software, algorithm | Bowtie | *Langmead et al., 2009* | RRID:SCR_005476 | |
| Software, algorithm | SAMtools | *Li et al., 2009* | RRID:SCR_002105 | |
| Software, algorithm | BEDtools | *Quinlan and Hall, 2010* | RRID:SCR_006646 | |
| Others | LSRII flow cytometer | BD Biosciences | RRID:SCR_002159 | |
| Others | FACSAria II Cell Sorter | BD Biosciences | RRID:SCR_018934 | |
| Others | Lionheart LX automated microscope | BioTex Instrument | RRID:SCR_019745 | |
| Others | 4-D Nucleofector | Lonza | NA | |

## Cell lines and cell culture

Abelson virus-transformed pre-B cells (abl pre-B cells) were generated as described previously (*Bredemeyer et al., 2006*). DNA Ligase 4-deficient (*Lig4*$^{-/-}$) abl pre-B cells were generated by deleting the LoxP site-flanked *Lig4* coding sequence by expressing Cre recombinase (*Helmink et al., 2011*). Deletion of the *Lig4* gene was verified by PCR. WT and *Lig4*$^{-/-}$ abl pre-B cells and MCF10A human mammary epithelial cells (from ATCC, CR-10317) used in this study all contain pCW-Cas9 (Addgene# 50661), which has a FLAG-tagged *Cas9* cDNA under the control of a doxycycline-inducible promoter. Intracellular staining with anti-FLAG and flow cytometry were used to identify clones homogeneously expressing FLAG-CAS9 after doxycycline treatment.

*Trp53bp1*$^{-/-}$, *Lin37*$^{-/-}$, *Lig4*$^{-/-}$:*Trp53bp1*$^{-/-}$, and *Lig4*$^{-/-}$:*Lin37*$^{-/-}$ cell lines were made by transiently transfecting *53* bp1 or *Lin37* gRNAs in the pX330 vector (Addgene# 42230) into WT or *Lig4*$^{-/-}$ cells followed by subcloning by limited dilution. Knockout clones were verified by western blot analysis for complete loss of expression of the proteins encoded by genes targeted by the gRNAs. Abl pre-B cells were cultured in Dulbecco's modified Eagle's medium (DMEM) supplemented with 10% fetal bovine serum (FBS) and 0.4% beta-mercaptoethanol, 100 U/ml penicillin/streptomycin, 1 mM sodium pyruvate, 2 mM L-glutamine, and 1× nonessential amino acids. MCF10A cells were cultured in DMEM/F12 supplemented with 5% horse serum, 20 ng/ml EGF, 0.5 μg/ml hydrocortisone, 100 ng/ml cholera toxin, 10 μg/ml insulin, and 100 U/ml penicillin-streptomycin.

To render cells non-cycling, abl pre-B cells were treated with 3 μM imatinib (Selleck Chemicals, S2475) for 2 (for chromatin-bound RPA assay) or 3 (for gene expression analysis) days and MCF10A cells were grown in EGF-free media for 2 days. At least two independent knockout clones of each gene in each cell type were generated and used in the experiments in this study. Knockout clones were verified by western blot analysis for lack of expression of the targeted proteins. MCF10A cell lines were authenticated by STR profiling, and MCF10A and murine cell lines tested negative for mycoplasma contamination.

## Chromatin-bound RPA assay

The chromatin-bound RPA flow cytometry assay was carried out as described previously with modifications (*Forment et al., 2012*). Briefly, the cells were washed with FACS wash (2% FBS in 1× phosphate-buffered saline [PBS]) followed by pre-extraction in Triton X-100 on ice for 10 min (0.05%

for imatinib-treated abl pre-B cells, 0.2% for proliferating abl pre-B cells, and 0.5% for MCF10A cells). The cells were washed again with FACS wash and fixed in BD Cytofix/Cytoperm at room temperature for 10 min. After fixation, the cells were incubated with anti-RPA32 antibody (Cell Signaling Technology, 2208S, 1:500) and anti-phospho-H2AX (S139) (Millipore Sigma, 05-636, 1:1000) in 1× BD Perm/Wash buffer at room temperature for 2 hr, followed by staining with Alexa Fluor 488 goat anti-rat IgG (BioLegend, 405418, 1:500) and Alexa Fluro 647 goat anti-moue IgG (BioLegend, 405322, 1:500) at room temperature for 1 hr. 20 μl of 7-AAD (BD Pharmingen, 559925) was added to each sample before resuspending cells in 300 μl of 1× PBS. RPA32 and phospho-H2AX (S139) levels were analyzed on a BD LSRFortessa Flow Cytometer.

For analysis of $G_1$ cells in a proliferating population, the cells were pulsed with 10 μM EdU for 1 hr prior to irradiation. Irradiated cells were kept in EdU-containing media during the course of the experiments and processed as described above. Following RPA32 and phosphor-H2AX(S139), click-IT chemistry was performed as per the manufacturer's instructions.

Non-cycling abl pre-B cells were exposed to 15 Gy of IR and analyzed 18 hr after irradiation. Cycling abl pre-B cells were analyzed 3 hr after 5 Gy of IR. Non-cycling MCF10A were exposed to 30 Gy of IR and analyzed 4 hr after irradiation. Cycling MCF10A cells were analyzed 6 hr after 25 Gy IR.

## Genome-wide guide RNA CRISPR/Cas9 screen

$Lig4^{-/-}$ or $Lig4^{-/-}:Lin37^{-/-}$ abl pre-B cells were transduced with lentiviral mouse genome-wide CRISPR gRNA library V2 (Addgene #67988) by centrifuging a cell and viral supernatant mixture (supplemented with 5 μg/ml polybrene) at 1800 rpm for 90 min. BFP-positive (stably transduced) cells were isolated on BD FACSAria II Cell Sorter, treated with 3 μg/ml doxycycline for 7 days followed by treatment with 3 μM imatinib for 2 days. 18 hr after exposing to 20 Gy IR, the cells were processed as described above for the chromatin-bound RPA flow cytometry assay and analyzed on a BD FACSAria II Cell Sorter. Cells with high (top 10%), low (bottom 10%) RPA staining, as well as unsorted cells were collected, and genomic DNA of these cells was harvested for amplification of gRNAs.

To generate an Illumina sequencing library, gRNAs in the selected cells were first amplified using primers pKLV lib330F and pKLV lib490R and the program '98°C/5 min - [98°C/15 s - 60°C/15 s - 68°C/1 min]×18–68°C/5 min'. The resulting PCR products were used as the template for additional PCR amplification using primers PE.P5_pKLV lib195 Fwd and P7 index180 Rev and the program '94°C/5 min - [94°C/15 s - 60°C/30 s - 68°C/20 s]×10–68°C/5 min' to add Illumina HiSeq adapters and indexes (RPI5:CACTGT, RPI6: ATTGGC, and RPI12: TACAAG). The final PCR products (~300 bp) were resolved in 1.5% agarose gel and purified by QIAQuick Gel Purification Kit (QIAGEN). Purified DNA was sequenced on Illumina HiSeq2500 system to determine gRNA representation in each sample (50 bp single-end reads).

Raw fastq files were demultiplexed by the Genomics and Epigenomics Core Facility of the Weill Cornell Medicine Core Laboratories Center. The gRNA sequence region was then retrieved from the sequencing data using Seqkit (*Shen et al., 2016*) and mapped to the gRNA sequence library (*Koike-Yusa et al., 2014*; *Tzelepis et al., 2016*). The number of reads of each library sequence was counted and then normalized as follows *Shalem et al., 2014*:

$$\text{Normalized reads per gRNA} = \frac{\text{reads per sgRNA}}{\text{total reads for all sgRNAs in sample}} \times 10^6 + 1$$

Finally, the enrichment score of a gRNA was calculated as a ratio of normalized reads of the gRNA in two samples.

pKLV lib330F: AATGGACTATCATATGCTTACCGT pKLV lib490R CCTACCGGTGGATGTGGAATG
PE.P5_pKLV lib195 Fwd: AATGATACGGCGACCACCGAGATCTGGCTTTATATATCTTGTGGAAAGGAC
P7 index180 Rev: CAAGCAGAAGACGGCATACGAGATINDEXGTGACTGGAGTTCAGACGTGTGCTCTTCCGATCCAGACTGCCTTGGGAAAAGC.

## Bulk gene inactivation

To inactivate *Trp53bp1*, *Lin37*, *Ctip*, *Brca1*, *Bard1*, *Blm*, or *Fancd2* in bulk cells populations, the cells were transduced with lentivirus pKLV-gRNAs by mixing cell suspensions with viral supernatant

supplemented with 5 µg/ml polybrene and 3 µg/ml doxycycline and spinning at 1800 rpm for 90 min. Stably transduced cells were sorted 3 days after transduction and sorted cells were kept in growth media with 3 µg/ml doxycycline for additional 24 hr before being subjected to analysis.

## Plasmid constructs

pCW-Cas9 was a gift from Eric Lander and David Sabatini (Addgene plasmid #50661) (*Wang et al., 2014*). pX330-U6-Chimeric_BB-CBh-hSpCas9 was a gift from Feng Zhang (Addgene plasmid #42230) (*Cong et al., 2013*). pKLV-U6gRNA(BbsI)-PGKpuro2ABFP was a gift from Kosuke Yusa (Addgene plasmid #50946) (*Koike-Yusa et al., 2014*). pLenti-CMV-Blast-PIP-FUCCI was a gift from Jean Cook (Addgene plasmid # 138715) (*Grant et al., 2018*). pHPRT-DR-GFP, pCBASceI, and pCBA were kindly provided by Dr. Maria Jasin (Memorial Sloan Kettering Cancer Center, New York, NY).

For LIN37 reconstitution experiments, Lin37 WT or CD mutant cDNAs were cloned into tetracycline-inducible TRE-Thy1.1 lentiviral vector to generate TRE-Lin37 (WT)-Thy1.1 or TRE-Lin37 (CD)-Thy1.1. WT Lin37 was amplified from Lin37 cDNA BC013546 (transOMIC) using primers Lin37 iso1_5'XhoI_S and Lin37 3'NotI_AS. To generate Lin37 CD mutant, Lin37 iso1_5'XhoI_S and Lin37 CD1_AS, and Lin37 CD2_S and 3'NotI_AS were first used to generate two cDNA fragments containing mutations in CD1 and CD2. The resulting PCR products were used in an overlapping PCR that annealed the two fragments to generate the full-length Lin37 (CD) mutant.

> Lin37 iso1_5'XhoI_S: GCCCTCGAGATGTTCCCGGTAAAGGTGAAAGTGG
> Lin37 3'NotI_AS: GCCGCGGCCGCTCACTGCCGGTCATACATCTCCCGT
> Lin37 CD1_AS: TACAGTGGTGTGTTCTCACTGAACTGGGCCAAGTCCACAGCCCCG GCAAA TAGCTTGATC
> Lin37 CD2_S: ACTTGGCCCAGTTCAGTGAGAACACACCACTGTACCCCATCGCCGG CGCC TGGATGCGCA

## End sequencing (End-seq)

End-seq was performed as previously described, using $20\times10^6$ abl pre-B cells harboring TET-inducible AsiSI-ER fusion treated with 3 µM imatinib (*Canela et al., 2016*). Briefly, the cells were embedded in agarose plugs, lysed, and treated with proteinase K and RNase A. The agarose-embedded genomic DNA was then blunted using ExoVII (NEB) and ExoT (NEB). Blunted DNA ends were A-tailed using Klenow exo- (NEB), and a biotinylated hairpin adaptor BU1 was ligated. After adaptor ligation, DNA was recovered after plug melting and treatment with beta-agarase. DNA was sheared to a length between 150 and 200 bp by sonication (Covaris), and biotinylated DNA fragments were purified using streptavidin beads (MyOne C1, Invitrogen). Following streptavidin capture, the newly generated ends were end-repaired using T4 DNA polymerase, Klenow fragment, and T4 polynucleotide kinase; A-tailed with Klenow exo-fragment (15 U); and finally ligated to hairpin adaptor BU2 using the NEB Quick Ligation Kit. After the second adaptor ligation, libraries were prepared by first digesting the hairpins on both adapters with USER enzyme (NEB) then PCR amplified for 16 cycles using TruSeq index adapters. All libraries were quantified using qPCR. Sequencing was performed on the Illumina NextSeq500 (75 bp single-end reads).

End-seq reads were aligned to the mouse genome (GRCm38p2/mm10) using Bowtie v1.1.2 with parameters (-n 3 k 1 l 50) and alignment files were generated and sorted using SAMtools and BEDtools (*Li et al., 2009*; *Quinlan and Hall, 2010*; *Langmead et al., 2009*). Heatmap was plotted using heatmap.2 of gplots package in R.

> BU1: 5'-Phos-GATCGGAAGAGCGTCGTGTAGGGAAAGAGTGUU[Biotin-dT]U[Biotin-dT]UUA-CACTCTTTCCCTACACGACGCTCTTCCGATC*T-3' [*phosphorothioate bond].
> BU2: 5'-Phos-GATCGGAAGAGCACACGTCUUUUUUUUAGACGTGTGCTCTTCCGA TC*T-3' [*phosphorothioate bond].

## Antibodies for western blot analysis

The following antibodies were used for western blot analysis: 53BP1 (Bethyl Laboratories, A300-272A, 1:3000), LIN37 (Santa Cruz Biotechnology, sc-515686, 1:200), BLM (Bethyl Laboratories, A300-572A, 1:2000), BRCA1 for mouse (R and D Systems, gift from Dr. Andre Nussenzweig, NCI, 1:1000) (*Zong et al., 2019*), BRCA1 for human (Millipore Sigma, 07-434, 1:1000), RAD51 (Millipore

Sigma, ABE257, 1:2000), BARD1 (Thermo Fisher Scientific, PA5-85707, 1:1000), CtIP (gift from Dr. Richard Baer, [Columbia University, New York], 1:1000), MRE11 (Novus Biologicals, NB100-142, 1:2000), RIF1 (Abcam, ab13422, 1:500), SHLD1/C20orf196 (Thermo Fisher Scientific, PA5-559280, 1:200), GAPDH (Sigma, G8795, 1:10,000), KAP1 (Genetex, GTX102226, 1:2000), FANCD2 (R and D Systems, MAB93691, 1:1000), BRCA2 for human (Proteintech, 19791-1-AP, 1:500), Rb1 (Thermo Fisher Scientific, LF-MA0173, 1:1000), Phospho-Rb (Ser780) (Cell Signaling Technology, 8180T, 1:1000), Phospho-Rb (Ser807/811) (Cell Signaling Technology, 8516T, 1:1000), PCNA (Bethyl Laboratories, A300-276A, 1:3000), CDK4: (Novus Biologicals, NBP1-31308, 1:1000), CDK4 (phosphor Thr 172) (GeneTex, GTX00778, 1:1000), and RPA (Cell Signaling Technology, 2208S, 1:1000).

## RNA- sequencing (RNA-Seq) analysis

RNA was purified from cycling $Lig4^{-/-}$ or $Lig4^{-/-}:Lin37^{-/-}$ cells treated with imatinib for 72 hr (two biological replicates each) using an RNeasy Mini Kit (QIAGEN). RNA-Seq libraries were prepared and directional RNA sequencing of $2\times50$ bp was performed at the Transcriptional Regulation and Expression Facility at Cornell University using NextSeq 500 sequencer. The raw fastq reads were first processed with Trim-Galore (Babraham Institute). The filtered reads were then aligned to GRCm38 reference genome with ENSEMBL annotations using Spliced Transcripts Alignment 2.7 (STAR 2.7) (*Dobin et al., 2013*). Differential expression was computed using DESeq2 (Bioconductor) and a false discovery rate 0.05 cutoff was used to identify sets of differentially expressed genes (*Love et al., 2014*).

For RNA-Seq of $G_1$ cells in proliferating cultures, $G_1$-phase cells ($Lig4^{-/-}$ or $Lig4^{-/-}:Lin37^{-/-}$ cells, two biological replicates) were isolated using PIP-FUCCI system (see below). RNA was prepared using the RNeasy Mini Kit (QIAGEN). RNA-Seq libraries were prepared and directional RNA sequencing of $2\times75$ bp was performed at the Genomics Core Libraries of Heflin Center of Genomic Sciences at University of Alabama at Birmingham using NextSeq 500 sequencer. RNA-Seq data was analyzed as described above.

## Ionizing radiation-induced foci formation assay

$Lig4^{-/-}$, $Lig4^{-/-}:Lin37^{-/-}$, and $Lig4^{-/-}:Trp53bp1^{-/-}$ abl pre-B cells were treated or not with 3 µM imatinib for 48 hr. Thereafter, the cells were pulsed with 10 µM EdU (Invitrogen) for 30 min and subjected to 10 Gy irradiation. For the detection of RAD51 foci, irradiated cells were allowed to recover for 4 or 20 hr, at which point they were immobilized on slides pre-coated with CellTak (Corning) and briefly pre-extracted (20 mM HEPES, 50 mM NaCl, 3 mM $MgCl_2$, 0.3 M sucrose, and 0.2% Triton X-100) on ice for 15 s to remove soluble nuclear proteins. Extracted samples were then fixed (4% paraformaldehyde), permeabilized (0.5% Triton X-100 in PBS), incubated with anti-RAD51 primary antibody (Abcam, ab176458, 1:250). Alternatively, irradiated (10 Gy) cells were allowed to recover for 1 hr prior to fixation without a preceding pre-extraction step, and subsequently incubated with primary antibodies recognizing 53BP1 (Novus Biologicals, NB100-305, 1:1000) or RIF1 (gift from Davide Robbiani [Rockefeller University, New York], 1:5000). In all cases, IRIFs were visualized by incubating samples with Alexa Fluor 555-conjugated secondary antibodies (Invitrogen). Where indicated, click-IT chemistry was performed as per the manufacturer's instructions. Finally, DNA was counterstained with DAPI (Thermo Fisher Scientific). Immunofluorescence images were captured at $40\times$ magnification on a Lionheart LX automated microscope (BioTek Instruments, Inc). Quantification of IRIF was performed using the Gen5 spot analysis software (BioTek Instruments, Inc).

## DR-GFP assay

To monitor HR activity using the DR-GFP reporter in non-cycling abl pre-B cells, the cells were first treated with 3 µM imatinib for 48 hr. 4 µg of pHPRT-DR-GFP, 4 µg of pCBASceI plasmids (or pCBA, the control plasmid without I-SceI) and 2 µg of pKLV-hCD2 (for monitoring transfection efficiency) were mixed with 10 million of imatinib-treated abl pre-B cells in 100 µl of Nucleofection Buffer SG (Lonza). The cell-DNA mixtures were pulsed in 4D Nucleofector X Unit using pulse code CM-147. Pulsed cells were placed in growth with 3 µM imatinib and cultured for 24 before analyzing the percentage of $GFP^+$ and $hCD2^+$ cells by flow cytometry. The percentages of $GFP^+$ cells between different samples were normalized by the percentages of $hCD2^+$ cells to take into account the difference of transfection efficiency among the samples.

## Guide RNA

Bulk gene inactivation or stable knockout mutants was achieved by CRSPR/Cas9 using the following gRNAs: gTrp53bp1 (GAACCTGTCAGACCCG ATC), gLin37 (AAGCTATTTGACCGGAGTG), gCtip (A TTAACCGGCTACGA AAGA), gBrca1 (GTCTACATTGAACTAGGTA), gBard1 (AAATCGTAAAGGCT GCCAC), gBlm (GATTTAACGAAGGAATCGG), gFancd2 (TCTTGTGATGTC GCTCGAC), gTrp53bp1 (human) (TCTAGTGTGTTAGATCAGG), and gLin37 (human) (TCTAGGGAGCGTCTGGATG).

## Cell cycle phase purification by PIP-FUCCI

Abl pre-B cells or MCF10A cells were transduced with pLenti-CMV-Blast-PIP-FUCCI and selected in 5 mg/ml Blasticidin for 3 days (*Grant et al., 2018*). To collect $G_1$-phase cells from proliferating cultures, mVenus-positive cells that were also mCherry-negative were sorted. To collect S/$G_2$/M cells, all mCherry-positive cells were sorted. Cell sorting was conducted on a BD FACSAria (BD Biosciences) at the Comprehensive Flow Cytometry Core at University of Alabama at Birmingham (supported by NIH P30 AR048311 and NIH P30 AI27667).

# Acknowledgements

The authors thank Dr. Maria Jasin for kindly providing plasmids for the DR-GFP assay. The authors thank UAB Comprehensive Flow Cytometry Core (National Institutes of Health grants P30 AI27667 and P30 CA013148) for assisting cell sorting and UAB Heflin Center for Genomic Sciences (National Institutes of Health grant P30 CA013148) and Transcriptional Regulation and Expression Facility at Cornell University for assisting (RNA Seq and data analysis). BPS is supported by the National Institutes of Health grants R01 AI047829 and R01 AI074953. JKT is supported by the National Institutes of Health grants R01 CA095641 and R01 GM064475. BPS and JKT are also supported by the Starr Cancer Consortium.

# Additional information

## Competing interests

Jessica K Tyler: Senior editor, *eLife*. The other authors declare that no competing interests exist.

## Funding

| Funder | Grant reference number | Author |
|---|---|---|
| National Institute of Allergy and Infectious Diseases | R01 AI047829 | Barry P Sleckman |
| National Institute of Allergy and Infectious Diseases | R01 AI074953 | Barry P Sleckman |
| National Cancer Institute | R01 CA095641 | Jessica K Tyler |
| National Institute of General Medical Sciences | R01 GM064475 | Jessica K Tyler |
| Starr Foundation | Starr Cancer Consortium | Jessica K Tyler Barry P Sleckman |
| National Institutes of Health | P30 AI27667 | Barry P Sleckman |
| National Institutes of Health | P30 CA013148 | Barry P Sleckman |

The funders had no role in study design, data collection and interpretation, or the decision to submit the work for publication.

## Author contributions

Bo-Ruei Chen, Conceptualization, Data curation, Formal analysis, Validation, Investigation, Visualization, Methodology, Writing - original draft, Project administration, Writing - review and editing; Yinan Wang, Anthony Tubbs, Dali Zong, Faith C Fowler, Nicholas Zolnerowich, Wei Wu, Formal analysis, Investigation, Writing - review and editing; Amelia Bennett, Chun-Chin Chen, Wendy Feng,

Investigation, Writing - review and editing; Andre Nussenzweig, Conceptualization, Supervision, Investigation, Writing - review and editing; Jessica K Tyler, Conceptualization, Resources, Supervision, Funding acquisition, Writing - original draft, Writing - review and editing; Barry P Sleckman, Conceptualization, Resources, Data curation, Supervision, Funding acquisition, Writing - original draft, Writing - review and editing

### Author ORCIDs
Bo-Ruei Chen ⬤ https://orcid.org/0000-0001-6404-2099
Wendy Feng ⬤ http://orcid.org/0000-0002-9734-8809
Jessica K Tyler ⬤ https://orcid.org/0000-0001-9765-1659
Barry P Sleckman ⬤ https://orcid.org/0000-0001-8295-4462

### Decision letter and Author response
Decision letter https://doi.org/10.7554/eLife.68466.sa1
Author response https://doi.org/10.7554/eLife.68466.sa2

## Additional files

### Supplementary files
• Source data 1. Compiled PDF file that contains raw images with the regions shown in the manuscript labeled.

• Source data 2. Raw images from *Figures 2* and *3*.

• Source data 3. Raw images from *Figure 4*.

• Source data 4. Raw images from *Figure 5*.

• Source data 5. Raw images from *Figure 5*.

• Source data 6. Raw images from *Figures 6* and *7*.

• Transparent reporting form

### Data availability
The original data for two genome-scale CRISPR/Cas9 screen and two RNA seq are included in the manuscript submission as figure supplemental source data or figure table supplements.

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
