## [Decision Letter]

**Acceptance summary:**

This manuscript will be of interest for scientists interested in cell cycle, DNA repair, transcription and genome stability opening a new chapter in studies of cell cycle dependent regulation of DSB repair. Much of the prior work focused on cell cycle driven post-translational regulatory modification of DSB end resection, whereas the current work finds transcriptional programs are equally, if not more important in controlling resection in G0. This could open possibilities for gene therapy in postmitotic tissues. The data are of high quality and the conclusions drawn are supported by the experimental evidence.

**Decision letter after peer review:**

Thank you for submitting your article "Lin37-DREAM Prevents DNA End Resection and Homologous Recombination at DNA Double Strand Breaks in Quiescent Cells" for consideration by *eLife*. Your article has been reviewed by 2 peer reviewers, and the evaluation has been overseen by a Reviewing Editor and Maureen Murphy as the Senior Editor. The reviewers have opted to remain anonymous.

Essential revisions

1) The authors show an ~2-fold increase in homology-mediated DSB repair in non-cycling Lig4-/- :Lin37-/- 347 abl-pre-B cells as compared to Lig4-/- abl-pre-B cells (Figure 6C) using an HPRT-DR-GFP reporter for DSB repair by HR. The consequences of having HR occurring in quiescent cells is not addressed.

In the minimum, the authors should add a paragraph discussing the biological significance of HR in quiescent cells. Does this recombination occur intrachromosomally or between homologs since no sister chromatid should be present? (See point #2)

The authors may want to consider expanding this point with additional experimental evidence by analyzing Lin37-deficient cells treated with IR in the quiescent state and then returned to cycling. One would expect genome instability and increased cell death to occur under these conditions if HR repair was active in G0. This addition is not essential for the revision.

2) The authors conclude that LIN37 functions to prevent DNA end resection primarily in quiescent G0 cells. It would be important to fully establish that Lin37-deficient cells still really are quiescent? Figure 2C shows 1% of noncycling Lin37-/- cells have BrdU uptake, whereas WT and 53BP1-/- are 0% in the Lig4-/- background.

3) Figure 6B should include data for no IR treatment so that the effects of IR treatment on RAD51 foci can be observed compared to untreated samples.

4) It is not shown whether or not Lin37-DREAM are localized on the genes that are dysregulated in Lin37-deficient cells. Additionally, since these effects do not occur in G1 but only in quiescence, is there any evidence that these genes are occupied by Lin37-DREAM only in G0 and not G1? While the current data support the conclusions drawn, such experimental additions would further strengthen the manuscript. New experiments are not essential for the revision, but the authors should perform GO analysis to identify which pathways are dysregulated by Lin37 loss (Figure 5B) and discuss whether these results are consistent with what is known about the gene cohort regulated by DREAM.

---

## [Author Response]

Essential revisions1) The authors show an ~2-fold increase in homology-mediated DSB repair in non-cycling Lig4-/- :Lin37-/- 347 abl-pre-B cells as compared to Lig4-/- abl-pre-B cells (Figure 6C) using an HPRT-DR-GFP reporter for DSB repair by HR. The consequences of having HR occurring in quiescent cells is not addressed.In the minimum, the authors should add a paragraph discussing the biological significance of HR in quiescent cells. Does this recombination occur intrachromosomally or between homologs since no sister chromatid should be present? (See point #2)The authors may want to consider expanding this point with additional experimental evidence by analyzing Lin37-deficient cells treated with IR in the quiescent state and then returned to cycling. One would expect genome instability and increased cell death to occur under these conditions if HR repair was active in G0. This addition is not essential for the revision.

As pointed out by the reviewer, in non-cycling LIN37-deficient cells, HR and aberrant

homology mediated joining occurs in the absence of sister chromatids. We feel that in G_0_ cells DNA ends that are resected will be resistant to repair by NHEJ and will be repaired by aberrant homology mediated mechanisms making use of homologous regions found on the same or other chromosomes. This will lead to intrachromosomal or interchromosomal joining, respectively, resulting in chromosomal deletions, inversions and translocations. In the revised manuscript we have discussed this point in more detail (lines 529-538, page 26).

2) The authors conclude that LIN37 functions to prevent DNA end resection primarily in quiescent G0 cells. It would be important to fully establish that Lin37-deficient cells still really are quiescent? Figure 2C shows 1% of noncycling Lin37-/- cells have BrdU uptake, whereas WT and 53BP1-/- are 0% in the Lig4-/- background.

Imatinib treatment of abl pre-B cells or EGF deprivation on MCF10A cells, results in the

majority of cells ceasing cell cycle in a 2N DNA state and stopping DNA replication as

evidenced by lack of BrdU incorporation (Figures. 2D, 2G and Figure 2—figure supplement 1B). In addition, CDK4 and RB phosphorylation, both hallmarks of the entry of quiescent cells into cycle, are both undetectable in Imatinib treated abl pre-B cells or EGF deprivation on MCF10A cells indicating that the vast majority of these cells at a G_0_ quiescent state (Figures 7A and 7B). We cannot rule out that a small fraction of cells (<1%) has entered the cell cycle, but these cells would not contribute significantly to our analyses of bulk populations where the majority of cells are quiescent.

3) Figure 6B should include data for no IR treatment so that the effects of IR treatment on RAD51 foci can be observed compared to untreated samples.

In the revised manuscript this data has been included in Figure 6B.

4) It is not shown whether or not Lin37-DREAM are localized on the genes that are dysregulated in Lin37-deficient cells. Additionally, since these effects do not occur in G1 but only in quiescence, is there any evidence that these genes are occupied by Lin37-DREAM only in G0 and not G1? While the current data support the conclusions drawn, such experimental additions would further strengthen the manuscript. New experiments are not essential for the revision, but the authors should perform GO analysis to identify which pathways are dysregulated by Lin37 loss (Figure 5B) and discuss whether these results are consistent with what is known about the gene cohort regulated by DREAM.

Gene Ontology (GO) analysis has been performed and is included in Figure 5-source data 1. A summary graph is now shown as Figure 5—figure supplement 1A. The over-represented GO groups of up-regulated genes in Lig4^-/-^:Lin37^-/-^ abl pre-B cells treated with imatinib from our RNA Seq show strong similarity to previous studies (e.g Mages et al., eLife (2017) ;6:e26876 and Litovchick et al., Molecular Cell (2007) 26, 539–551). This includes genes encoding proteins that function in cell cycle, cell division, chromosome segregation, DNA repair and cellular responses to DNA damage. While we did not perform ChIP seq analysis, many genes up-regulated in imatinib-treated Lig4^-/-^:Lin37^-/-^ abl pre-B cells have been shown to be bound by the DREAM complex in the aforementioned two published studies.

To determine whether the LIN37-DREAM complex only functions in G_1_ phase cells, we

carried out RNA seq analysis on G_1_ phase cells isolated from proliferating cultures using the PIP-FUCCI system as described in experiments shown in Figures 7C and 7D. There are about 500 genes that are up-regulated in G_0_ or G_1_-phase Lig4^-/-^:Lin37^-/-^ abl pre B cells, compared to Lig4-/- abl pre-B cells (298 in G_0_ cells and 361 in G_1_ cells). Among these genes, 145 genes are uniquely up-regulated in G_0_ cells, 208 genes in G_1_-phase cells and 153 genes in both cell populations. Interestingly, genes up-regulated uniquely in G_0_ or in both G_0_ and G_1_ phase both show over-representation in GO terms of cell cycle regulation and DNA repair. This result suggests that in addition to its function during quiescence, the LIN37-DREAM complex can also negatively regulate the expression of a subset of genes encoding proteins that function in cell cycle regulation and DNA replication in G1 phase cells. These new data are now shown Figure 7-table supplement 1 and 2 and discussed in the Results Section (lines 406-423, pages 20 and 21).

Although these new data indicate that LIN37-DREAM can function in both quiescence and G1 phase of the cell cycle, we show that the protein levels of some of the LIN37-DREAM targets are increased in quiescent, but not in G1 phase, LIN37-deficient cells (Figures 5C, 5D, 7C, 7D and Figure 5—figure supplement 1B). These results suggests that in proliferating cells at G1-phase, additional mechanisms must regulate the activity of proteins that function in DNA end processing and HR.